# Multiplex, single-cell CRISPRa screening for cell type specific regulatory elements

Florence M. Chardon [1,2,10], Troy A. McDiarmid[1,2,10], Nicholas F. Page[3,4,5], Riza M. Daza [1,2], Beth K. Martin[1,2], Silvia Domcke[1], Samuel G. Regalado[1], Jean-Benoît Lalanne[1], Diego Calderon [1], Xiaoyi Li[1,2], Lea M. Starita [1,6], Stephan J. Sanders[3,5,7], Nadav Ahituv [4,5] ✉ & Jay Shendure [1,2,6,8,9] ✉

CRISPR-based gene activation (CRISPRa) is a strategy for upregulating gene expression by targeting promoters or enhancers in a tissue/cell-type specific manner. Here, we describe an experimental framework that combines highly multiplexed perturbations with single-cell RNA sequencing (sc-RNA-seq) to identify cell-type-specific, CRISPRa-responsive *cis*-regulatory elements and the gene(s) they regulate. Random combinations of many gRNAs are introduced to each of many cells, which are then profiled and partitioned into test and control groups to test for effect(s) of CRISPRa perturbations of both enhancers and promoters on the expression of neighboring genes. Applying this method to a library of 493 gRNAs targeting candidate *cis*-regulatory elements in both K562 cells and iPSC-derived excitatory neurons, we identify gRNAs capable of specifically upregulating intended target genes and no other neighboring genes within 1 Mb, including gRNAs yielding upregulation of six autism spectrum disorder (ASD) and neurodevelopmental disorder (NDD) risk genes in neurons. A consistent pattern is that the responsiveness of individual enhancers to CRISPRa is restricted by cell type, implying a dependency on either chromatin landscape and/or additional *trans*-acting factors for successful gene activation. The approach outlined here may facilitate large-scale screens for gRNAs that activate genes in a cell type-specific manner.

There are millions of candidate *cis*-regulatory elements (cCREs) in the human genome, yet only a handful have been functionally validated and confidently linked to their target gene(s)[1]. Recently, we and others have combined CRISPR-interference (CRISPRi) and sc-RNA-seq to scalably validate distal cCREs, while also linking them to the gene(s) that they regulate[1–4]. However, to date, the vast majority of work in the field has focused on screening candidate regulatory elements for *necessity*, with only a few studies screening for *sufficiency* in the endogenous context.

CRISPR-activation (CRISPRa) is a versatile approach that allows one to test for the sufficiency of cCRE activity[5–8]. CRISPRa screens of noncoding regulatory elements have at least four potential advantages over CRISPRi screens. First, as noted above, CRISPRa can identify cCREs that are sufficient even if not singularly necessary to drive target

[1]Department of Genome Sciences, University of Washington, Seattle, WA, USA. [2]Seattle Hub for Synthetic Biology, Seattle, WA, USA. [3]Department of Psychiatry and Behavioral Sciences, Kavli Institute for Fundamental Neuroscience, Weill Institute for Neurosciences, University of California, San Francisco, San Francisco, CA, USA. [4]Department of Bioengineering and Therapeutic Sciences, University of California, San Francisco, San Francisco, CA, USA. [5]Institute for Human Genetics, University of California, San Francisco, San Francisco, CA, USA. [6]Brotman Baty Institute for Precision Medicine, Seattle, WA, USA. [7]Institute of Developmental and Regenerative Medicine, Department of Paediatrics, University of Oxford, Oxford OX3 7TY, UK. [8]Howard Hughes Medical Institute, Seattle, WA, USA. [9]Allen Discovery Center for Cell Lineage Tracing, Seattle, WA, USA. [10]These authors contributed equally: Florence M. Chardon, Troy A. McDiarmid. ✉e-mail: Nadav.Ahituv@ucsf.edu; shendure@uw.edu

gene expression. Second, CRISPRa can identify elements that, when targeted, may upregulate already active genes above their baseline levels. Third, CRISPRa has the potential to discover inactive regions that, when transcriptional activation machinery is recruited to them, can act as active enhancers and increase expression of nearby genes[9]. Finally, CRISPRa has the potential to identify cCRE-targeting gRNAs whose activity is cell type-specific, opening the door to "cis regulatory therapy" (CRT) for haploinsufficient and other low-dosage associated disorders, as recently demonstrated for monogenic forms of obesity and autism spectrum disorder[10,11]. However, despite these potential advantages, CRISPRa targeting of noncoding regulatory elements has mostly been deployed in an ad hoc manner[9,12–14], meaning typically focusing on a single target gene, and furthermore in workhorse cancer cell lines rather than more physiologically relevant in vitro models.

Here, we present a scalable framework in which we introduce multiple, random combinations of CRISPRa perturbations to each of many cells followed by sc-RNA-seq (Fig. 1), analogous to an approach that we previously developed for CRISPRi screening[2,4]. Computational partitioning of cells into test and control groups based on detected gRNAs enables greater power than single-plex CRISPRa screens, as any given single-cell transcriptome is informative with respect to multiple perturbations[2]. In this proof-of-concept study, we performed two screens in which the same set of cCREs was targeted, first in K562 cells and then in human iPSC-derived excitatory neurons. We discover both enhancer and promoter-targeting gRNAs capable of mediating upregulation of target gene(s). For enhancers in particular, the upregulatory potential of individual gRNAs was consistently restricted to one cell type, implying a dependency on either the cis chromatin landscape and/or additional trans-acting factors for successful gene activation.

## Results

### Multiplex single-cell CRISPRa screening of regulatory elements in K562 cells

As a proof of principle, we first sought to implement multiplex single-cell CRISPRa screening in the chronic myelogenous leukemia cell line K562, an ENCODE Tier 1 cell line[15] in which we had previously performed a multiplex CRISPRi screen[2]. Our proof-of-concept library included gRNAs targeting transcription start site (TSS) positive

controls (30 gRNAs), candidate promoters (313 gRNAs), candidate enhancers (100 gRNAs) and non-targeting controls (NTCs; 50 gRNAs). The 30 TSS positive control gRNAs were selected from a previously reported hCRISPRa-v2 library[16], while the 313 candidate promoter-targeting gRNAs were designed to 50 annotated TSSs of 9 high-confidence haploinsufficient risk genes associated with ASD and NDD (BCL11A, TCF4, ANK2, CHD8, TBR1, SCN2A, SYNGAP1, FOXP1, and SHANK3)[17]. Genes for which haploinsufficient (i.e. dominant, loss-of-function) variants appeared to be primary drivers of risk were prioritized. Further prioritization was based on gene cDNA size being too large to fit into traditional gene therapy vectors. The candidate enhancer-targeting guides included 50 gRNAs designed to target 25 enhancer hits previously validated by CRISPRi[2], as well as 50 gRNAs designed to target 25 enhancer "non-hits" (i.e. sequences with biochemical markers strongly predictive of enhancer activity in K562 cells that did not alter gene expression when targeted with CRISPRi[2]) (Supplementary Fig. 1a, b; "Methods"). We cloned this gRNA library (n = 493) into piggyFlex, a piggyBac transposon-based gRNA expression vector, to allow for genomic integration and stable expression of gRNAs[18]. The piggyFlex vector has both antibiotic (puromycin) and fluorophore (GFP) markers, enabling flexibly stringent selection for cells with higher numbers of gRNA integrants. Transposon vectors also avoid issues that can arise due to recombination during viral packaging[19] and associated safety concerns. Additionally, this vector design allows for gRNA transcript capture during single-cell library preparation[18] (Supplementary Fig. 1c).

There is no consensus on which CRISPRa activation complex is best suited for broad and scalable targeting of enhancers[13]. We therefore tested both the VP64 activation complex, which consists of four copies of the VP16 effector, and the VPR activation complex, which consists of the VP64 effector fused to the p65 and Rta effectors[20,21]. We generated a monoclonal, stably VP64-expressing K562 cell line, purchased a polyclonal, stably VPR-expressing K562 cell line (Fig. 2a; "Methods"), and validated the capacity of these lines for CRISPRa with a minimal cytomegalovirus (CMV) promoter-tdTomato reporter expression assay[22] (Supplementary Fig. 2).

We then transfected the gRNA library and piggyBac transposase into each cell line at a 20:1 library-to-transposase ratio to achieve high

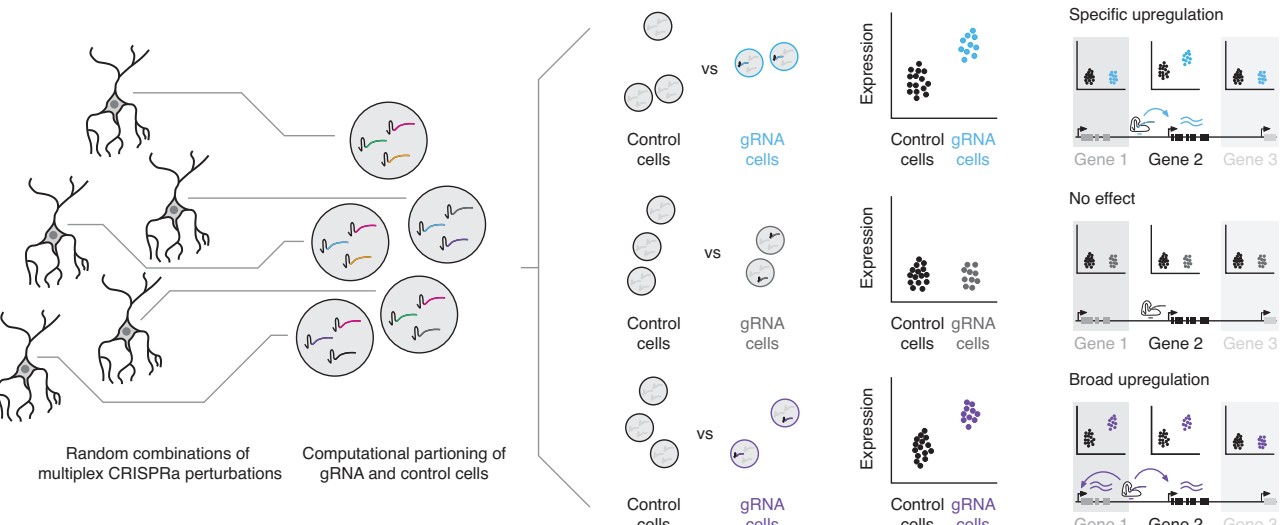

**Fig. 1 | Multiplex, single cell CRISPRa screening for cell type-specific regulatory elements.** (Left) A library of gRNAs targeting candidate cis-regulatory elements (cCREs) is introduced in a multiplex fashion to a population of cells expressing CRISPRa machinery, such that each cell contains a random combination of multiple CRISPRa-mediated perturbations. (Middle) Following single cell transcriptional profiling and gRNA assignment, cells are systematically computationally partitioned into those with or without a given gRNA and tested for upregulation of neighboring genes. (Right) CRISPRa perturbations can either result in target-specific upregulation, no detectable effect (e.g., for non-targeting controls) or, at least theoretically, broad cis-upregulation of multiple genes in the vicinity of the gRNA/CRISPRa machinery. Furthermore, patterns of upregulation can either be general or cell type-specific.

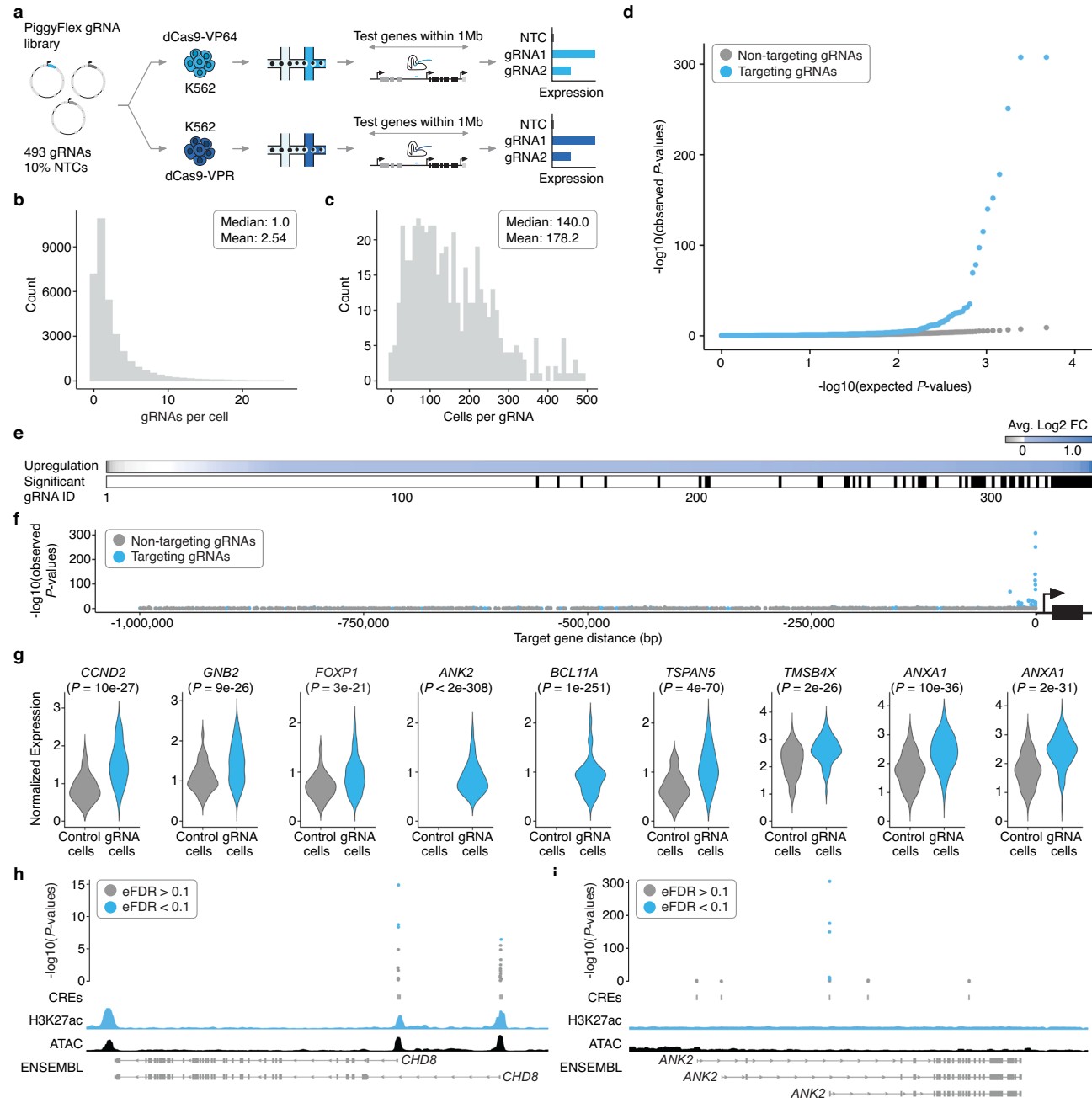

**Fig. 2 | Multiplex single cell CRISPRa screening of regulatory elements in K562 cells. a** Screen workflow. **b** gRNAs/cell. **c** Cells/gRNA. **d** Quantile-quantile plot showing distribution of expected vs. observed *P*-values for targeting (blue) and non-targeting (gray, downsampled) differential expression tests. *P*-values are from a two-tailed Wilcoxon rank-sum test. **e** (Top) Average log2 fold change in expression between cells with each targeting gRNA vs. controls for each of the primary/programmed target genes. Tests are sorted left-to-right by increasing log2 fold change. (Bottom) Categorical heatmap showing which of the perturbations drove significant upregulation using an Empirical FDR approach (EFDR < 0.1). **f** Targeting gRNAs yielding significant upregulation are enriched for proximity to their target gene. We observe no such enrichment for NTCs tested for associations with target genes randomly selected from the same set. **g** Average log2 fold change between cells with a given gRNA and controls for select hit gRNAs. Number of cells bearing each targeting gRNA (from left to right): *CCND2* (*n* = 73), *GNB2* (*n* = 220), *FOXP1* (*n* = 313), *ANK2*

(*n* = 403), *BCL11A* (*n* = 191), *TSPAN5* (*n* = 48), *TMSB4X* (*n* = 260), *ANXA1* (*n* = 166), *ANXA1* (*n* = 128). Control cells are downsampled to have the same number of cells as the average number of cells detected per gRNA (*n* = 178) for visualization. Normalized expression values represent log normalized expression values from Seurat. Only cells with at least 1 target gene UMI are plotted. Note that some genes (e.g., *ANK2*) are typically not detected as expressed in this cell context, resulting in zero UMIs detected and thus no expression distribution plotted in downsampled control cell populations. *P*-values as in panel (**d**, **h**) Hits included multiple gRNAs targeting isoform-specific promoters of *CHD8*. *P*-values are visualized alongside tracks for K562 ATAC-seq (ENCODE), H3K27ac signal (ENCODE), and RefSeq validated transcripts (ENSEMBL/NCBI). *P*-values as in panel **d** EFDR sets as in panel (**e**). **i** The strongest hit gRNAs for ANK2 target the same promoter that is not prioritized by biochemical marks (e.g., accessibility or H3K27ac). Genomic tracks, *P*-values, and EFDR sets as in panel **h** Abbreviations: NTC non-targeting controls.

multiplicity of integration (MOI), and selected cells with puromycin. Cells were cultured for nine days before harvesting for sc-RNA-seq to capture and assign gRNAs to single cell transcriptomes (Fig. 2a; Supplementary Fig. 1). After QC filtering, we recovered 33,944 high-quality single-cell transcriptomes across the two cell lines, with 79% of cells having one or more detected gRNAs. We recovered a mean of 2.5 gRNAs per cell (Fig. 2b) and 178 cells per gRNA (Fig. 2c). Transcriptome quality, MOI, gRNA assignment rate, and gRNA coverage were similar across all four sc-RNA-seq batches (10x Genomics lanes) as well as the two cell lines tested (Supplementary Fig. 3).

To systematically assess the effect of each CRISPR perturbation on target gene expression, we adapted an iterative differential expression testing strategy in which all single cell transcriptomes are computationally partitioned into cells with or without a given gRNA[2]. These two groups are then tested for differential expression of all genes within 1 megabase (Mb) (upper estimate of topologically associated domain size in mammalian genomes[23]) upstream and downstream of the gRNA target site (Fig. 1; Fig. 2a; "Methods"). In both VP64- and VPR-mediated CRISPRa screening experiments, we observed clear upregulation from both promoter and enhancer-targeting gRNAs (276/391 $\log_2FC > 0$, 70.6%, $P < 2.2 \times 10^{-16}$, Fisher's Exact Test; Fig. 2d, e). The presence of an excess of highly significant $P$-values for cells harboring targeting gRNAs versus non-targeting controls (NTCs) also indicates that this multiplex framework successfully detects upregulation of genes from CRISPRa perturbations (Fig. 2d). Effects were consistently much stronger and more significant in the dCas9-VP64 cell line as compared to the dCas9-VPR line (Supplementary Fig. 3). This may be due to differences between the VP64 and VPR effectors, site-of-integration effects (VP64 line is monoclonal while VPR line is polyclonal), MOI differences of the integrated effectors, power differences (more cells were recovered per perturbation for the VP64 line than the VPR line), or a combination of these factors.

To identify significant associations between cCRE-targeting gRNAs and their target genes, which we term "hit gRNAs", we set an empirical false discovery rate (FDR) threshold based on the $P$-values from the NTC gRNA differential expression tests, which are subject to the same sources of noise and error as the targeting gRNA tests. Using an empirical FDR cutoff of 0.1 ("Methods"), we identified 59 activating gRNA hits, including 8 TSS-targeting positive control gRNAs, 39 candidate promoter-targeting gRNAs, 9 distal enhancer hit gRNAs, 2 distal enhancer hit gRNAs wherein the target gene of CRISPRa vs. CRISPRi differed, and 1 distal enhancer non-hit gRNA (in the last three contexts, hit vs. non-hit refers to whether they were "hits" in the previous CRISPRi-based screen with the same guides and cell line[2]) (Fig. 2e; Supplementary Fig. 4). Successfully activating gRNAs were strongly enriched for targeting regions proximal to the genes that they upregulated (Fig. 2f) and were specific to their predicted target (45/47 promoter-targeting gRNA hits and 9/12 successful enhancer-targeting gRNAs exclusively upregulated the predicted target and no other gene within 1 Mb; Supplementary Fig. 4; Supplementary Data 2–4). The gRNAs that upregulated a gene other than the predicted target are discussed further below. Of note, we also observed no instances where targeting a regulatory element, whether a promoter or enhancer, caused significant upregulation of >1 gene.

Taken together, these results demonstrate the potential of this framework to efficiently identify promoter- or enhancer-targeting gRNAs that drive specific upregulation of their target genes in a cell type of interest. Of note, the promoters that were successfully targeted with CRISPRa included genes that were already well-expressed (e.g., *CCND2*, *GNB1*), including two that are haploinsufficient neurodevelopmental disease genes (*FOXP1*, *CHD8*) (Fig. 2g, h; Supplementary Fig. 4; Supplementary Fig. 5; Supplementary Data 2–4). For *CHD8*, in which variants leading to haploinsufficiency are important risk factors for ASD and NDD[24,25], we identified multiple CRISPRa gRNAs targeting distinct isoform-specific promoters (Fig. 2h; Supplementary Fig. 4).

Our strongest hits were at the promoters of genes with very low or undetectable expression in K562 (e.g., *ANK2*, *BCL11A*; Fig. 2g; Supplementary Fig. 5; Supplementary Data 2–4). For example, we identified multiple CRISPRa gRNAs targeting *ANK2*, an ASD/NDD risk gene with a complex isoform structure[24,25] that is very lowly expressed in K562 cells (Fig. 2i). Interestingly, the strongest hits for *ANK2* all targeted a TSS that is not prioritized by biochemical marks (i.e., it is relatively inaccessible and displays a low degree of H3K27ac in K562 cells compared to candidate TSSs of other genes in our library; Fig. 2i). On the other hand, for many targeted TSSs or promoters, only one gRNA, if any, activated their target gene when coupled to CRISPRa. More specifically, out of the 313 candidate promoter-targeting gRNAs designed to 50 annotated TSSs of 9 genes, only 37 gRNAs, targeting 11 TSSs and 5 genes, successfully mediated upregulation. One gRNA upregulated a different gene (*WWC3*) than the intended target (*FOXP1*). These results underscore the value of inclusive, empirical screens to identify both CRISPRa-competent promoters as well as gRNAs that can successfully activate them.

At the outset of this work, it was unclear if targeting CRISPRa perturbations to enhancers alone (without co-targeting putatively associated promoters) could reliably increase target gene expression to an extent detectable with conventional sc-RNA-seq[9,12,13]. To determine if CRISPRa targeted to a single enhancer alone could effectively upregulate target gene expression, we analyzed our 50 targeted candidate enhancers, 25 of which were previously validated by multiplex CRISPRi in K562 cells[2]. We observed target gene upregulation for 8 of these 50 targeted candidate enhancers (as noted above, mediated by 12 gRNAs; Fig. 2g; Supplementary Fig. 4; Supplementary Fig. 5). Six of the 8 enhancers come from the set of 25 enhancer-gene pairs that we also identified with CRISPRi[2], including several cases where distinct gRNAs targeting the same enhancer are both successful, e.g. two CRISPRa-competent enhancers of *ANXA1* (Fig. 2g; Supplementary Fig. 4; Supplementary Fig. 5). In addition, we identified: (1) an enhancer-targeting gRNA that was not a hit in the CRISPRi screen, but here led to upregulation of *HMGA1*; and (2) two enhancer-targeting gRNAs that mediate downregulation of *TUBA1A* when coupled to CRISPRi, but upregulation of *ASIC1* when coupled to CRISPRa.

*ASIC1* is not typically detected as expressed in K562 cells, explaining why e-ASIC1 was not detected as necessary for *ASIC1* expression via CRISPRi. In other words, this enhancer was detected as necessary for high *TUBA1A* expression (an enhancer-gene pair discovered with CRISPRi) but sufficient for increasing *ASIC1* expression when targeted with CRISPRa. While the precise mechanisms underlying this differential behavior of e-ASIC1 in relation to these two genes remains to be characterized, it may be that be that *ASIC1* requires a higher degree of stimulation (in this case, introduced via a CRISPRa activation), whereas *TUBA1A* is activated sufficiently via the same enhancer at a lower level of stimulation (i.e. at the baseline resting state of K562 cells). Alternatively, the *TUBA1A* link may be a false positive of the CRISPRi study[2], which used an empirical 10% FDR for identifying hits.

Taken together, these results show that multiplex CRISPRa screens leveraging sc-RNA-seq can identify enhancer-targeting gRNAs that can mediate upregulation of specific genes without co-targeting of the corresponding promoters (Fig. 2g; Supplementary Fig. 4; Supplementary Fig. 5; Data 2–4). Furthermore, differences in activity and target-choice despite using the same gRNAs hint at potential differences between CRISPRi and CRISPRa that warrant further exploration.

## Multiplex single-cell CRISPRa screening of regulatory elements in post-mitotic iPSC-derived neurons

We next sought to extend this framework beyond K562 cells to a model that is more physiologically relevant, post-mitotic human induced pluripotent stem cell (iPSC)-derived neurons (Fig. 3a)[26]. We initially attempted to generate an iPSC line with CRISPRa-VP64 machinery

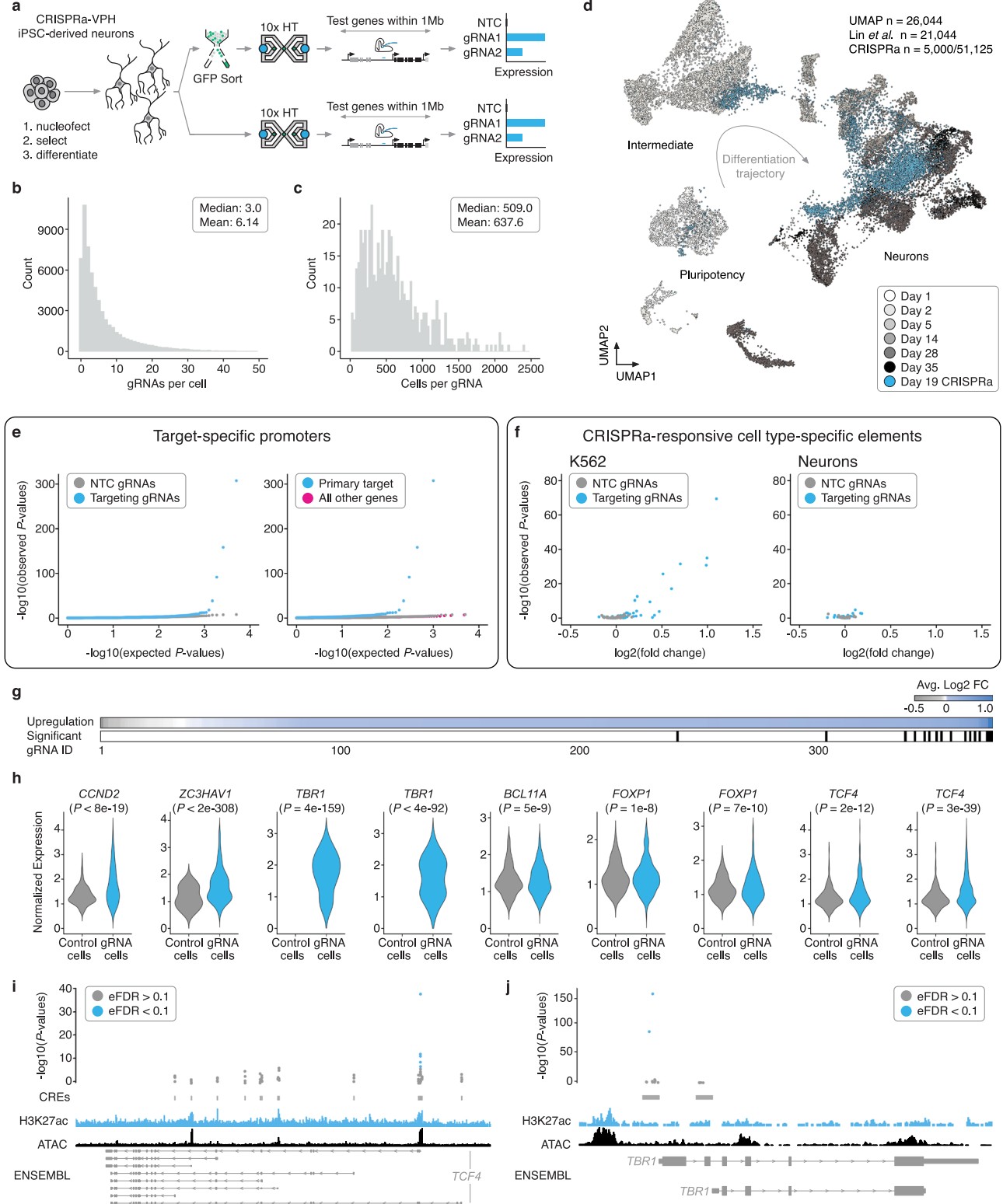

integrated randomly via lentiviral transduction. However, monoclonal lines generated with this approach silenced the CRISPRa-VP64 machinery during neural differentiation, preventing use in our screening framework. Indeed, delivery of CRISPRa machinery to post-mitotic neurons is considerably more challenging than workhorse cancer cell lines and requires more complex cell engineering and delivery approaches[27]. To circumvent this, we used a WTC11 iPSC line equipped with a doxycycline-inducible *NGN2* transgene expressed

from the *AAVS1* safe-harbor locus to drive neural differentiation, as well as an ecDHFR-dCas9-VPH construct, expressed from the *CLYBL* safe-harbor locus, to drive CRISPRa (Supplementary Fig. 6A, B)[6]. In this line, addition of doxycycline to induce *NGN2* expression and tri-methoprim (TMP) to inhibit the ecDHFR degrons drives neural dif-ferentiation and initiates CRISPRa[6]. Expression of NGN2 in iPSCs commits these cells to a neuronal fate, and post-mitotic neurons with neuronal morphology develop within days[28].

**Fig. 3 | Multiplex single cell CRISPRa screening of regulatory elements in post-mitotic iPSC-derived neurons. a** Screen workflow. **b** gRNAs/cell. **c** Cells/gRNA. **d** UMAP projection of the neuron dataset from this study (blue, 51,183 cells downsampled to 5000 cells to aid with visualization) onto a sc-RNA-seq differentiation time-course from a similar differentiation protocol and NGN2 iPSC line (21,044 cells)[29]. **e** (Left) QQ-plot displaying observed vs. expected $P$-value distributions for targeting (blue) and NTC (downsampled) populations. (Right) QQ-plot for targeting tests against their intended/programmed target (blue) compared to targeting tests of all other genes with 1 Mb of each gRNA (pink) and NTCs (gray downsampled). $P$-values are from a two-tailed Wilcoxon rank-sum test. **f** Average log2 fold change and $P$-values exclusively for gRNAs that target putative enhancers in K562 cells (left) and iPSC-derived neurons (right). $P$-values as in panel (**e**). **g** (Top) Average log2 fold change in expression between cells with each targeting gRNA vs. controls for each of the primary/programmed target genes. (Bottom) Categorical heatmap showing which of the perturbations produced significant upregulation using an Empirical FDR approach (EFDR < 0.1). **h** Average log2 fold change between cells with a given gRNA and controls for select hit gRNAs (plotted as in Fig. 2d). Number of cells bearing each targeting gRNA (from left to right): $CCND2$ ($n = 350$), $ZC3HAV1$ ($n = 765$), $TBR1$ ($n = 455$), $TBR1$ ($n = 655$), $BCL11A$ ($n = 964$), $FOXP1$ ($n = 1198$), $FOXP1$ ($n = 917$), $TCF4$ ($n = 1051$), $TCF4$ ($n = 1253$). Control cells are downsampled to have the same number of cells as the average number of cells detected per gRNA ($n = 638$) for visualization. $P$-values as in (**e**). **i** Of 14 targeted candidate promoters, five hit gRNAs for TCF4 target the same candidate promoter that aligns with biochemical marks of regulatory activity (ATAC-Seq and H3K27ac). Empirical $P$-values are visualized alongside tracks for iPSC-derived neuron ATAC-seq (accessibility)[54], and H3K27ac[54], and RefSeq validated transcripts (ENSEMBL/NCBI). $P$-values as in panel (**e**). EFDR sets as in panel (**g**). **j** Hits included multiple gRNAs targeting $TBR1$. Genomic tracks, $P$-values, and EFDR sets as in panel (**i**).

After optimizing iPSC transfection conditions to achieve high numbers of integrated gRNAs per cell via nucleofection, we integrated the same gRNA library (at a 5:1 gRNA-library:transposase ratio) into iPSCs as we did for the K562 screen (Fig. 3a). Following integration, we confirmed functional CRISPRa activity in these neurons via the same tdTomato expression assay used in our K562 CRISPRa validation (Supplementary Fig. 6b). In addition to optimizing transfection conditions, we sought to further boost the multiplicity of gRNA integrations per cell by selecting the cells with a high concentration of puromycin (Fig. 3a). After differentiating to neurons over 19 days, we proceeded to sc-RNA-seq. Half of the neurons went directly into sc-RNA-seq (10x Genomics), while the other half were dissociated and flow sorted based on GFP expression (top 40%) prior to sc-RNA-seq, again with the goal of boosting the multiplicity of gRNA integrations (Fig. 3a). After quality control filtering, we retained 51,183 single-cell transcriptomes, of which we recovered 1+ associated gRNAs for 89%. With our optimized transfection protocol, we identified a mean of 6.1 gRNAs/cell (Fig. 3b) and a mean of 638 cells that harbored each individual gRNA (Fig. 3c). Sorting on GFP expression prior to sc-RNA-seq resulted in a 2-fold increase in the number of gRNAs identified per cell (Supplementary Fig. 7).

Our differentiated neurons most closely resemble 14- to 35-day differentiated neurons obtained via $NGN2$ induction in iPSCs by an independent group[29] (inferred by integration of these sc-RNA-seq datasets; Fig. 3d; Supplementary Fig. 8). A minority of the neurons transcriptionally resemble an intermediate neuronal fate, a difference that we tentatively attribute to the absence of co-cultured glia in our differentiation protocol. Although glia are known to promote maturation of NGN2-induced neurons (and were used in generating the dataset we are comparing to[29]), we excluded them because they can also introduce culture variability due to batch effects introduced by primary glia[28].

We confirmed that the neurons had progressed beyond a pluripotent state and were committed to a post-mitotic neuronal fate by the expression of the pan-neuronal marker $MAP2$ and the lack of expression of the pluripotency marker $NANOG$ (Supplementary Fig. 8). These neurons also express $LHX9$ and $GPM6A$, markers of central nervous system (CNS) neurons (Supplementary Fig. 8c), and $CUX1$ and $SLC17A7$, but not GABAergic markers $GAD1$ and $GAD2$, supporting their assignment as excitatory rather than inhibitory neurons (Supplementary Fig. 8f)[26]. Consistent with this, when we co-embedded our transcriptome data onto data from Lin et al.[29], they overlay with "Fate 2" and "Fate 3" cells, which transcriptionally resemble CNS neurons (Fig. 3d; "Methods"). Of note, there was no readily apparent enrichment of specific gRNAs within particular clusters (Supplementary Fig. 9), which is consistent with the specificity and modest fold changes of the observed instances of upregulation (Supplementary Fig. 9).

We applied the same differential expression testing strategy as used for the K562 screen to the iPSC-derived neuron screen data, with an empirical FDR cutoff of 0.1 to call significant hits. Similarly to the K562 screen, we observed clear upregulation from targeting gRNAs (281/383 $\log_2$FC > 0, 73.4%, $P < 2.2 \times 10^{-16}$, Fisher's Exact Test) and an excess of highly significant $P$-values for targeting gRNA tests compared to NTCs (Fig. 3e), confirming that this overall framework is transferable to more physiologically and clinically relevant models such as iPSC-derived neurons. As with the K562 screen, we observed strong enrichment for genomic proximity between successful gRNAs and their target genes, but no such enrichment for NTCs tested for associations with target genes randomly selected from the same set (Supplementary Fig. 10).

There were 17 hit gRNAs in neurons (FDR < 0.1; Fig. 3g), all of which were TSS-targeting positive controls ($n = 6$) or candidate promoters of ASD/NDD risk genes ($n = 11$) (Supplementary Fig. 11). Of these 17 hit gRNAs, 12 were also hits in the K562 screen while 5 were specific to iPSC-derived neurons (Supplementary Fig. 12a). The screen in iPSC-derived neurons was strikingly target-specific: 16 of 17 of our identified hits, all promoter-targeting gRNAs, upregulated their anticipated target gene and no other genes within the 1-Mb window tested (Supplementary Data 5–7). The only gRNA hit in iPSC-derived neurons resulting in upregulation of an unintended gene was a gRNA targeting the TSS of the pseudogene $PPP5D1$ that led to upregulation of the calmodulin gene $CALM3$ (Supplementary Fig. 11d), but this is presumably due to these two genes sharing a bidirectional, outward-oriented core promoter. This gRNA also drove upregulation of $CALM3$ in the CRISPRa screen of K562 cells (Supplementary Fig. 4d). These results, in combination with the K562 results discussed earlier, reveal that certain gRNAs designated 'TSS-targeting positive controls' based on their ability to yield a growth phenotype in bulk screens of cancer cell lines, do not yield strong upregulation of their putative target gene (Supplementary Fig. 4c; Supplementary Fig. 11c). We observed no significant differences across several characteristics (e.g., GC content, baseline target gene expression level, the number of cells harboring each gRNA) between gRNAs yielding successful activation and those not in K562 cells and neurons, with the exception that K562 enhancer hit gRNAs tended to have more cells (Supplementary Fig. 13). However, upon conducting an analysis of epigenetic features available for both cell lines as well as for in vivo developing brain samples, we found that promoter- and enhancer- targeting hit gRNAs were more likely to fall within regions of active, open chromatin and were conversely depleted from regions with repressive marks (Supplementary Fig. 14; Supplementary Data 8).

Similar to K562 cells, we observed several instances where a specific TSS was most amenable to activation (Supplementary Fig. 15). One such example is $TCF4$, an ASD/NDD risk gene[24,25]. We tested 14 candidate TSSs of $TCF4$ and identified 5 gRNAs capable of driving upregulation of $TCF4$ in neurons, all of which target the same candidate TSS that resides in open chromatin with strong H3K27ac signal (Fig. 3h, i; Supplementary Figs. 15a, 16). Our hits also included examples of cell

type-specific promoters. Among these were several gRNAs targeting candidate promoters of ASD/NDD risk genes capable of upregulating genes that are not expressed or rarely expressed in iPSC-derived *NGN2*-differentiated neurons (Fig. 3h; Supplementary Fig. 16). For example, gRNAs targeting the promoter of *TBR1*, a transcription factor expressed in forebrain cortical neurons but known not to be expressed in NGN2-differentiated iPSC-derived neurons[26] led to *TBR1* upregulation (Fig. 3j; Supplementary Figs. 15b, 16). Of note, these same gRNAs did not result in upregulation of *TBR1* in K562 cells. This suggests that these neurons are in a permissive state for CRISPRa to activate *TBR1*, despite a lack of highly accessible chromatin in the region targeted by the *TBR1* gRNA (Fig. 3h,j; Supplementary Fig. 15b). Whether these differences in "*TBR1* activatability" are due to differences in the chromatin environment at this locus between K562 cells and iPSC-derived neurons, or alternatively differences in the milieu of *trans*-acting factors, remains an open question.

However, in contrast to the cell type-specific promoter examples noted above, we more often observed consistent upregulation across promoter targets and TSS-targeting controls between the two cell types (Supplementary Fig. 12). Specifically, 12 out of 17 of the promoter- and TSS-targeting hit gRNAs in neurons were also hits in K562 cells, and upregulation was correlated across cellular contexts (Pearson's correlation coefficient = 0.75, Supplementary Fig. 12). In contrast, we observed striking cell type-specificity for targeted enhancers that were successfully upregulated. While 20% (12/60) of our K562 screening hits were enhancer-targeting gRNAs (Supplementary Fig. 4), none of these were also hits in neurons (Supplementary Fig. 11; Supplementary Fig. 17). Even putting aside significance, the fold-effects on the anticipated target genes of K562-competent activating gRNAs were not well-correlated between cell types (Fig. 3f; Supplementary Fig. 12b, Pearson's correlation coefficient = −0.18). Overall, these results show that it is possible to drive cell type-specific upregulation of a gene of interest by targeting CRISPRa to a cell type-specific distal enhancer, without co-targeting of the corresponding promoter.

## Reanalysis with covariates and singleton experimental validations support and extend results

To better account for covariates, we applied SCEPTRE[30–33], an independently developed analytical framework based on conditional resampling that integrates various covariates to calibrate the statistical assessment of results in single cell CRISPR screens ("Methods"). SCEPTRE detected clear signals in both datasets (Supplementary Figs. 18, 19; Supplementary Data 9–14). Encouragingly, SCEPTRE results aligned well with our original approach, but yielded more hits. Our original approach identified 59 activating hit gRNAs in K562 cells and 17 activating hit gRNAs in neurons, while SCEPTRE found 83 K562 hits and 35 neuron hits. 48/59 K562 (81%) and 15/17 neuron (88%) hits found with our approach were also found by SCEPTRE (Supplementary Fig. 18, 19; Supplementary Data 9–14). The few cases where our original approach found a hit that SCEPTRE did not, were primarily due to the fact SCEPTRE has a more stringent minimum threshold on the proportion of cells required to show expression of the target gene (e.g., *ASIC1* expression is rarely detected in K562 cells, yet two independent gRNAs targeting a distal enhancer led to clear upregulation in our screen; see also singleton validations below).

Several notable findings arose from this analysis. First, SCEPTRE identified a single hit gRNA yielding upregulation of *SCN2A* exclusively in iPSC-derived neurons out of 20 *SCN2A*-targeting gRNAs tested (Supplementary Fig. 17; Supplementary Data 14). This is the same "H1 gRNA" we recently used in rescue studies in *SCN2A*+/− haploinsufficient mice and human iPSC-derived neurons[11]. Second, despite more than double the hits in neurons (35 neuron hits with SCEPTRE vs. 17 with our original approach), and several new enhancer hits in K562 cells (21 enhancer hits with SCEPTRE vs. 12 with our original approach),

the signal from K562 enhancers is still overwhelmingly cell type-specific (SCEPTRE calls only 1 enhancer hit in neurons vs. 21 enhancer hits in K562 cells) (Supplementary Figs. 18, 19; Supplementary Data 9–14).

To further validate our results, we selected 8 of our hit gRNAs to test in singleton experiments. Validation gRNAs were selected to represent a range of significance levels, as well as both shared and cell-type specific promoters and enhancers (Supplementary Data 15). We cloned the 8 gRNAs as singleton constructs, introduced them at high MOI to create 3 independent polyclonal cell lines in each cell context, selected, and in the case of neurons, differentiated cells, and then created bulk RNA-seq libraries for each line (8 gRNAs × 2 cell contexts × 3 replicate cell lines = 48 independent polyclonal lines and corresponding bulk RNA-seq samples). We then compared expression levels in the 3 replicate lines with a given targeting gRNA to expression levels in the remaining 21 lines from that cell context as a control (mirroring our single-cell analysis framework).

Our validation set included sgRNA hits that are expected to drive upregulation in both cell types (*BCL11A, DNMT3B, TCF4, FOXP1, HMGA1*), in K562 cells only (*ANXA1, ASIC1*), or in iPSC-derived neurons only (*TBR1*). In 6/7 (K562) or 5/6 cases (iPSC-derived neurons), the expected upregulation was observed, validation rates consistent with the 0.1 FDR of the initial screens (Fig. 4; Supplementary Fig. 20). Further, gRNAs targeting a cell-type specific promoter of *TBR1* drove the expected upregulation exclusively in iPSC-derived neurons (Fig. 4). Hit gRNAs targeting e-ANXA1 and e-ASIC1, two enhancers identified as K562-specific in our single-cell analysis, drove upregulation of *ANXA1* and *ASIC1* exclusively in K562 cells in singleton validations (Fig. 4). Of note, e-HMGA1, a putative shared enhancer, drove the expected upregulation in neurons but not in K562 in singleton validations, so we removed it from our list of high-confidence, singleton validated CRISPRa-responsive CREs (Fig. 4). These experiments were designed to validate upregulations of target genes in targeted tests, though in some cases we were able to detect additional biologically coherent gene expression changes on a genome-wide basis, potentially secondary effects of upregulating the primary target (Supplementary Figs. S21, S22, Supplementary Data 16). For example, in both K562 cells and iPSC-derived neurons, genes that were significantly downregulated upon CRISPRa of *BCL11A* are highly enriched for genes involved in oxidative phosphorylation (Gene Set Enrichment Analysis (GSEA); FDR *q*-value < 1e-20). Importantly, fold changes in target gene expression observed in singleton bulk RNA-seq validation experiments correlated well with results from our multiplex single-cell framework (Pearson's correlation coefficient = 0.83) (Supplementary Fig. 23). Taken together, these results refine and extend our framework and confirm our ability to identify target- and cell type- specific CRISPRa-responsive regulatory elements and gRNAs that target them.

## Discussion

Here we describe a scalable, multiplex single-cell CRISPRa screening framework to identify cell type-specific CREs, the genes they regulate, and gRNAs that successfully target them. In applying this framework, we identified gRNAs functionally and cell type-specifically targeting endogenous CREs of haploinsufficient genes in K562 cells and iPSC-derived excitatory neurons. To our knowledge, this is the first screen of this scale and kind (highly multiplexed, single-cell resolution noncoding CRISPRa perturbations) performed successfully in a post-mitotic, iPSC-derived human cell type. Our results include several examples of enhancer-target gene relationships that are experimentally supported in their native genomic context, which remain exceedingly rare in the literature due to the significant technical barriers that have persisted despite major efforts to discover or confirm more such relationships. Importantly, our approach lays the groundwork for discovering hundreds to thousands more endogenous enhancer-gene relationships in non-workhorse cancer

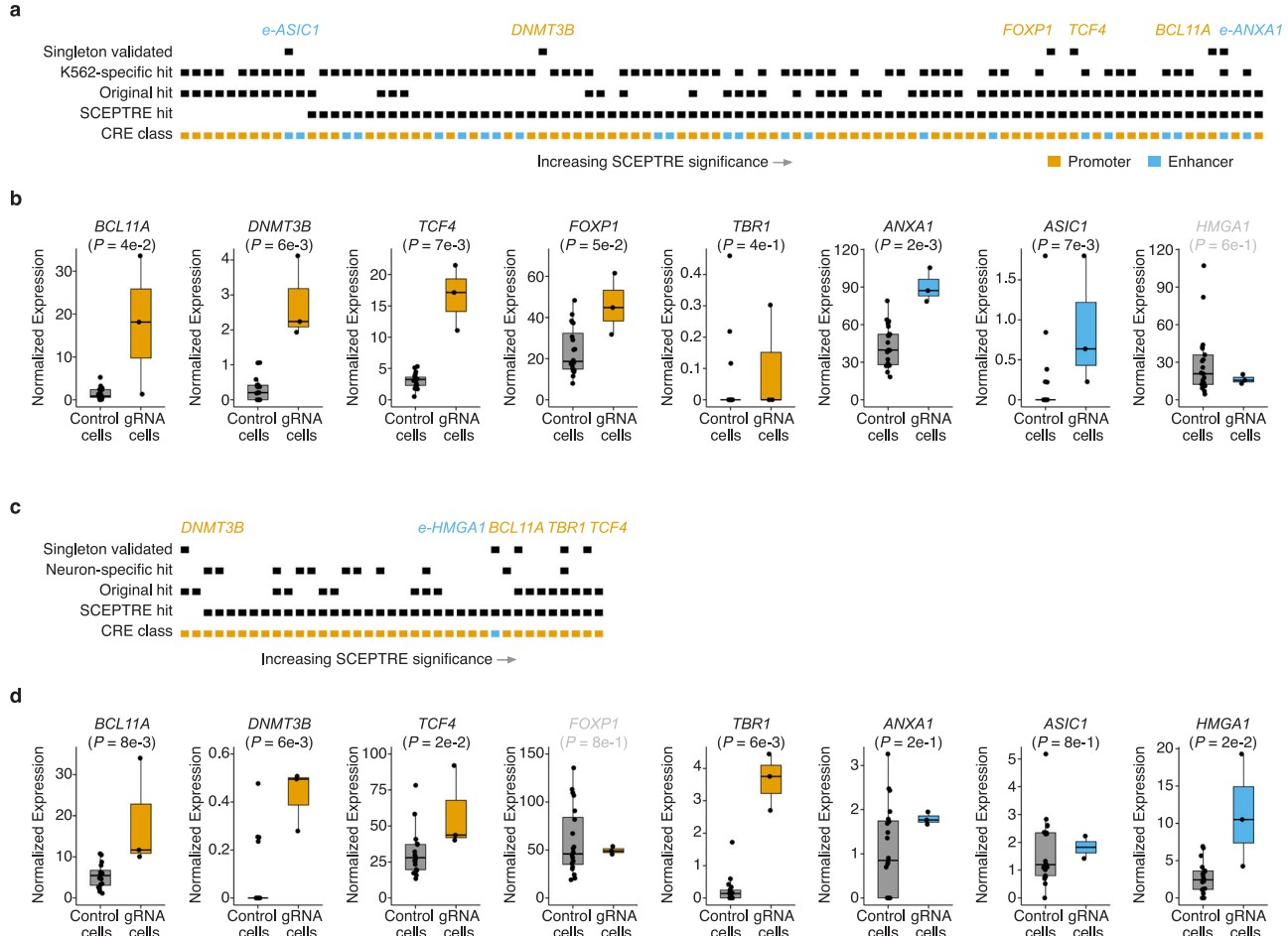

**Fig. 4 | Reanalysis with covariates and singleton experimental validations support and extend results. a** Singleton validation results for K562 cells. Categorical heatmap indicating whether a K562 hit gRNA was detected with SCEPTRE, our original approach, targeted a promoter or enhancer, drove K562-specific upregulation, and whether it was validated with singleton experiments. Hit CREs that drove upregulation in singleton validations are labeled and coloured according to target CRE class. **b** Boxplots displaying the average log2 fold change between cells with a given gRNA ($n = 3$ independent lines) and controls ($n = 21$ lines). Normalized expression displayed in transcripts per million (TPM). Boxes represent the 25th, 50th, and 75th percentiles. Whiskers extend from hinge to 1.5 times the inter-quartile range. All data points are also plotted on top of the box plot for transparency. *P*-values are from a two-tailed Wilcoxon rank-sum test with a significance threshold of 0.1. **c** Singleton validation results for iPSC-derived neurons.

Categorical heatmap indicating whether a neuron hit gRNA was detected with SCEPTRE, our original approach, targeted a promoter or enhancer, drove iPSC-derived neuron-specific upregulation, and whether it was validated with singleton experiments. Hit CREs that drove upregulation in singleton validations are labeled and coloured according to target CRE class. **d** Boxplots displaying the average log2 fold change between cells with a given gRNA ($n = 3$ independent lines) and controls ($n = 21$ lines). Normalized expression displayed in transcripts per million (TPM). *P*-values as in panel (**b**). The two instances where expected upregulation was not observed in a particular cell context are labeled in gray above the corresponding box plot (specifically gRNAs targeting e-HMGA1 in K562 cells and a promoter of *FOXP1* in neurons). Boxes as in panel (**d**). The 14/16 experiments where the expected result was observed are labeled in black above the corresponding box plot, while the 2/16 that are labeled in gray failed to validate.

cell lines such as post-mitotic iPSC-derived neurons. For example, we anticipate it will be possible to massively scale screening for gRNAs and cell-type-specific CREs capable of upregulating remaining functional copies of the roughly 660 genes known to cause disease when haploinsufficient.

In principle, increases in target gene expression of less than 100% to a few fold, especially if cell-type specific, would be ideal for therapeutic purposes in haploinsufficient disorders[10,11,17]. Many of our fold-changes meet these criteria (median fold-change for all hits = 1.7; 79/118 SCEPTRE hits achieved at least 50% upregulation; 42/118 hits achieved at least a 2-fold upregulation; 49/118 hits achieved between 1.5 and 2.5-fold upregulation). Furthermore, even in cases where upregulation is lower or higher than theoretically ideal at the mRNA level, there can still be therapeutic benefit (e.g., in cases where partial rescue of expression is sufficient for phenotypic improvement, or where post-transcriptional mechanisms buffer overcorrection of expression levels)[11,17].

Several of our strongest gRNA hits were not prioritized by typical predictors of enhancer function, such as chromatin accessibility or H3K27ac histone modifications. For example, we are able to upregulate *TBR1* in iPSC-derived neurons by targeting a promoter region that is largely within closed chromatin in this cellular context. Indeed, while measures of proximity, accessibility, and enhancer-related biochemical marks are all strong predictors, none are conclusive or deterministic predictors of regulatory sequence function, either alone or in combination. This underscores the importance of empirical, systematic screens for CRISPRa-responsive regulatory sequences with approaches such as the one described here. Ultimately, a variety of factors including chromatin accessibility and epigenetic modifications, gRNA design and target-specific nuances around CRISPRa-responsiveness, appear to collectively shape the likelihood and extent of success of a given CRISPRa perturbation in a given cellular context. Our framework has now provided a set of validated CRISPRa sgRNAs that could be used as positive controls moving forward in other

CRISPRa screens (in at least K562 and iPSC-derived postmitotic neurons; the set that overlap may be the best bet for potential 'universal' controls), thereby filling a critical gap. Future scaling of this technology and its application to additional, clinically relevant cell types, may provide rich training sets that may enable derivation of rational CRISPRa gRNA design rules for distal, cell-type-specific gene activation, which, in contrast to promoters and CRISPRi[16,34,35], are quite lacking at present. Further, these results illustrate the unique potential of noncoding CRISPRa screens to identify regulatory elements that can mediate upregulation of target genes, regardless of whether or not the gene is natively expressed in the cell type of interest or not.

A major question that we sought to answer through these experiments was whether one can target candidate enhancer sequences with a CRISPRa perturbation and observe upregulation of an intended target gene via scRNA-seq. There have been relatively few efforts to apply CRISPRa to enhancers to date, and most have focused on a handful of enhancer regions and typically only measuring changes in expression of a specific nearby gene of interest as a readout[9,12,13]. Recent literature suggests that co-targeting a promoter and the candidate enhancer in question can make the enhancer CRISPRa perturbations more efficient and reliable[13]. Although feasible, co-targeting an enhancer and promoter is less likely to yield cell-type-specific upregulation of target genes, is more complex and increases the chances of effects on off-target genes (not to mention off-target cell types). Despite using gRNAs that were optimized for CRISPRi screening in our CRISPRa screen, we observed target gene upregulation for 8 of 25 enhancers that we targeted (32%), showing that one can reliably increase target gene expression by targeting enhancers alone. We imagine that this success rate can be improved via a combination of brute force, i.e. testing more gRNAs, improved CRISPRa activation domains[36], and better CRISPRa-specific gRNA design.

Multiplex single-cell CRISPRa screening is a scalable approach to identifying functional CRISPRa gRNAs that can upregulate intended target genes in either a general or cell-type-specific manner. We introduced multiple perturbations per cell, which increased the power of our assay (i.e. a mean of 1 gRNA per cell would have required sc-RNA-seq of >400,000 cells to achieve the same power). For additional context, the increased hands-on time and reagents required to generate, select, and differentiate the 48 cell lines used to validate the 16 gRNAs across the two cell contexts made the costs of the singleton and pooled experiments comparable, despite singleton experiments assessing <5% as many gRNAs. Alternative bulk approaches such as those based on integrated fluorescent reporters or FISH scale linearly with a new set of reagents required for each new candidate target gene.

One potential concern is that multiplex, single-cell CRISPR screens that treat cells as pseudo-replicates have the potential to suffer from an overestimation of differentially expressed genes due to the statistical dependency between cells originating from the same starting population[37]. Our singleton validation results allay this concern, but we cannot fully rule out overestimation. Further, given the ease of generating large numbers of differentiated neurons with in vitro human neural cultures, and then sorting on the GFP-positive gRNA expression vector prior to single-cell transcriptome profiling, our approach offers a straightforward way to further boost the number of gRNAs captured per cell. In addition, improvements in methods to capture specific transcripts[38–40], or to perform scRNA-seq more cheaply[41,42], may enable considerably larger screens for a given cost.

## Methods
### Cell lines and culture
**K562 cell culture.** K562s cells (ATCC; Cat. No. CCL-243) are a pseudotriploid ENCODE Tier I erythroleukemia cell line derived from a female (age 53) with chronic myelogenous leukemia[15]. All K562 cells were grown at 37 °C, and cultured in RPMI 1640 + L-Glutamine (GIBCO, Cat. No. 11-875-093) supplemented with 10% fetal bovine serum (Fisher Scientific, Cat No. SH3039603) and 1% penicillin-streptomycin (Thermo Fisher Scientific, Cat. No. 15070063).

**Induced pluripotent stem cell (iPSC) culture.** Human WTC11 iPSCs equipped with a doxycycline-inducible *NGN2* transgene expressed from the *AAVS1* safe-harbor locus as well as an ecDHFR-dCas9-VPH construct (VPH consists of 12 copies of VP16, fused with a P65-HSF1 activator domain) expressed from the CLYBL safe-harbor locus were a gift from the Kampmann lab[6]. These iPSCs were cultured in mTeSR Plus Basal Medium (Stemcell technologies; Cat. No. 100-0276) on Greiner Cellstar plates (Sigma-Aldrich; assorted Cat. Nos.) coated with Geltrex™ LDEV-Free, hESC-Qualified, Reduced Growth Factor Basement Membrane Matrix (Gibco; Cat. No. A1413302) diluted 1:100 in Knockout DMEM (GIBCO/Thermo Fisher Scientific; Cat. No. 10829018). mTeSR Plus Basal Medium was replaced every other day. When 70–80% confluent, cells were passaged by aspirating media, washing with DPBS (GIBCO/Thermo Fisher Scientific; Cat. No. 14190144), incubating with StemPro Accutase Cell Dissociation Reagent (GIBCO/Thermo Fisher Scientific; Cat. No. A1110501) at 37 °C for 5 min, diluting Accutase 1:1 in mTeSR Plus Basal Medium, collecting cells in conical tubes, centrifuging at $800\,g$ for 3 min, aspirating supernatant, resuspending cell pellet in mTeSR Plus Basal Medium supplemented with 0.1% dihydrochloride ROCK Inhibitor (Stemcell technologies; Cat. No. Y-27632), counting and plating onto Geltrex-coated plates at the desired number.

**Human iPSC-derived neuronal cell culture, differentiation, and CRISPRa induction.** The iPSCs described above were used for the differentiation protocol below. On day −3, iPSCs were dissociated and centrifuged as above, and pelleted cells were resuspended in Pre-Differentiation Medium containing the following: Knockout DMEM/F-12 (GIBCO/Thermo Fisher Scientific; Cat. No. 12660012) as the base, 1X MEM Non-Essential Amino Acids (GIBCO/Thermo Fisher Scientific; Cat. No. 11140050), 1X N-2 Supplement (GIBCO/ Thermo Fisher Scientific; Cat. No. 17502048), 10 ng/mL NT-3 (PeproTech; Cat. No. 450-03), 10 ng/mL BDNF (PeproTech; Cat. No. 450-02), 1 ug/mL Laminin mouse protein (Thermo Fisher Scientific; Cat. No. 23017015), 10 nM ROCK inhibitor, and 2 mg/mL doxycycline hyclate (Sigma-Aldrich; Cat. No. D9891) to induce expression of *NGN2*. iPSCs were counted and plated at 800 K cells per Geltrex-coated well of a 12-well plate in 1 mL of Pre-Differentiation Medium, for three days. At day −2 and day −1, media changes were performed using pre-differentiation medium without ROCK inhibitor. On day −1, 12-well plates for differentiation were coated with 15 ug/mL Poly-L-Ornithine (Sigma-Aldrich; Cat. No. P3655) in DPBS, and incubated overnight at 37 degrees Celsius. On day 0, the Poly-L-Ornithine coated plates were washed three times using DPBS, and the plates were air dried in a 37 degree Celsius incubator until all the DPBS evaporated. Pre-differentiated cells were dissociated and centrifuged as above, and pelleted cells were resuspended in Maturation Medium containing the following: 50% Neurobasal-A medium (GIBCO/Thermo Fisher Scientific; Cat. No. 10888022) and 50% DMEM/F-12 (GIBCO/Thermo Fisher Scientific; Cat. No. 11320033) as the base, 1X MEM Non-Essential Amino Acids, 0.5X GlutaMAX Supplement (GIBCO/Thermo Fisher Scientific; Cat. No. 35050061), 0.5X N-2 Supplement, 0.5X B-27 Supplement (GIBCO/Thermo Fisher Scientific; Cat. No. 17504044), 10 ng/mL NT-3, 10 ng/mL BDNF, 1 ug/mL Laminin mouse protein, and 2 ug/mL doxycycline hyclate. Pre-differentiated cells were subsequently counted and plated at 400,000–450,000 cells per well of a 12-well plate coated with Poly-L-Ornithine in 1 mL of Maturation medium with 20 uM trimethoprim (TMP) (Sigma-Aldrich, Cat No. 92131) to activate the CRISPRa machinery in these cells (TMP stabilizes the degron-tagged CRISPRa machinery). On day 7, half of the medium was removed and an equal volume of fresh Maturation medium without doxycycline was added. On day 14, half of the medium was removed and twice that volume of

fresh medium without doxycycline was added. On day 19, neurons were harvested for sc-RNA-seq.

## Cell line generation and CRISPRa validation

**K562 cells**. K562 cells expressing dCas9-VP64 were generated in-house via lentiviral integration of a dCas9-VP64-blast construct[7] (Addgene Plasmid #61422) into K562 cells (ATCC; Cat. No. CCL-243). Cells were selected with 10 ug/mL blasticidin, and polyclonal cells were single-cell sorted into 96-well plates to grow up clonal cell lines expressing dCas9-VP64. Clonal cell lines were tested for CRISPRa activity by testing the ability of a CRISPRa gRNA to activate a minP-tdTomato construct[21], and the highest tdTomato expressing cell line was used for experiments. K562 cells expressing dCas9-VPR were purchased from Horizon Discovery/Perkin Elmer (Cat. No. HD dCas9-VPR-005), and these cells were tested for CRISPRa activity using the same tdTomato expression assay described above.

**iPSC-derived neurons**. Human WTC11 iPSCs equipped with a doxycycline-inducible *NGN2* transgene expressed from the *AAVS1* safe-harbor locus as well as an ecDHFR-dCas9-VPH construct expressed from the CLYBL safe-harbor locus were a gift from the Kampmann lab[6]. These cells were tested for CRISPRa activity using the same tdTomato expression assay that was used to validate the K562 cell lines, which is described above.

## gRNA selection and design

A complete breakdown of gRNA library contents and overview of the gRNA design pipeline is illustrated in Fig. S1. Briefly, enhancer-targeting gRNAs were selected from our CRISPRi library[2,43]. Specifically, 50 spacer sequences (2 per candidate enhancer) were randomly selected from the list of 664 significant "hit" enhancer-gene pairs in the at-scale library. Another 50 spacer sequences targeting an additional 25 candidate enhancers (again 2 per candidate enhancer) were randomly selected from candidate enhancer non-hits (i.e., gRNAs from the at-scale library targeting candidate enhancer regions with strong biochemical marks predictive of regulatory activity that did not yield significant downregulation of any neighboring genes in our previous CRISPRi study). An additional 30 TSS-positive control gRNAs were randomly sampled from the top quartile of gRNAs recommended by Horlbeck et al. (hCRISPRa-v2 library)[16]. 50 NTC negative control spacer sequences were also selected from the hCRISPRa-v2 library[16]. The 313 candidate promoter targeting gRNAs were either selected from the Horlbeck et al. library[16] or designed using FlashFry[43] (Fig. S1). Briefly, 50 candidate promoters of 9 NDD risk genes (*TCF4, FOXP1, SCN2A, CHD8, BCL11A, TBR1, SHANK3, SYNGAP1, ANK2*)[24,25] were pulled from Basic GENCODE annotations[44] and were filtered for "type" == "transcript" and "transcript_type" == "protein coding". Separate bed files were generated for all promoter regions defined as the 500 bp upstream of each protein coding transcript. Careful attention was paid to the strand orientation of each transcript when annotating promoter regions. Bed files were sorted and merged to combine multiple promoters with >1 bp overlap into a single promoter annotation. Transcript bounds provided for each merged promoter begin +1 bp from the end of the promoter and end at the position corresponding to the longest transcript mapping to that promoter. NGG-protospacer within these candidate promoters were identified using FlashFry and subsequently scored using default parameters (see FlashFry manuscript and user guide for a complete description of scoring metrics/algorithms)[43]. A TSS-distance metric was then calculated for each gRNA using human fetal brain 5′ Capped Analysis of Gene Expression (CAGE) data[45,46] obtained from FANTOM (https://fantom.gsc.riken.jp/5/sstar/FF:10085-102B4; CTSS, hg38). First, the strongest FANTOM annotated TSS was identified within each +/−500 bp region up and downstream of each hg38 Gencode Basic protein coding transcript TSS. For regions with a tie between the highest scoring FANTOM TSSs, the TSS position

closest to Gencode annotated TSS position was prioritized. Each candidate sgRNA from FlashFry was annotated with the distance to the nearest FANTOM TSS using the command "bedtools closest -a sgRNAs_with_fantom_tss -b strongest_fantom_tss_within_gencode_promoter -D b -t first." For Gencode Basic protein coding transcripts without a human fetal brain FANTOM peak within 500 +/− bp, the distance of each sgRNA to the nearest Gencode TSS was reported instead. A distance of zero indicates that an sgRNA overlaps with the nearest annotated TSS. Multiple rounds of successively relaxing score and distance thresholds were then iterated until the top 4 gRNAs for each candidate promoter were selected (five selection rounds in total). Optimal TSS-distances were approximated using genome-wide CRISPRa design rules[34]. gRNAs flagged for potentially problematic poly-thymidine tracks or GC content were excluded. The gRNA selection criteria used in each round were as follows:

**Round 1: 1**. TSS Distance between −150 and −75 BP **2**. Doench2014OnTarget >= 0.2 **3**. Dangerous_in_genome <= 1 **4**. Hsu2013 > 80.

**Round 2: 1**. TSS Distance between −400 and −50 BP **2**. Doench2014OnTarget >= 0.2 **3**. Dangerous_in_genome <= 1 **4**. Hsu2013 > 80.

**Round 3: 1**. TSS Distance between −400 and −50 BP **2**. Doench2014OnTarget >= 0.2 **3**. Dangerous_in_genome <= 1 **4**. Hsu2013 > 50.

**Round 4: 1**. TSS Distance between −400 and −50 BP 2. Doench2014OnTarget >= 0.2 3. Dangerous_in_genome <= 2 4. Hsu2013 > 50.

**Round 5: 1**. Doench2014OnTarget >= 0.2 **2**. Dangerous_in_genome <= 2 **3**. Hsu2013 > 10 **4**. DoenchCFD_maxOT <0.95

Complete oligo sequences with gRNA spacers and additional sequences for cloning into piggyFlex are listed in Supplementary Data 1. Note all gRNAs in our library are designed/modified to start with a G followed by the 19 base pair spacer to facilitate Pol III transcription.

## gRNA library cloning into piggyFlex vector

The 493 gRNAs with associated 10 N random barcodes were ordered as an IDT oPool and PCR amplified with Q5 High-Fidelity polymerase (NEB, Cat. No. M0491S) to make double stranded DNA. The piggyFlex backbone vector was digested with SalI (NEB, Cat. No. R3138S) and BbsI (NEB, Cat. No. R0539S) in 10X NEBuffer r2 at 37 degrees Celsius overnight to ensure complete digestion of the backbone. This digestion cuts out the EF1a-puro-GFP cassette of the vector which is then added back in a later cloning step. The digestion product was run on a 1% agarose gel in TAE buffer, and the linear backbone vector (5098 base pairs in size) was gel extracted using a gel extraction kit (NEB, Cat. No. T1020S). The second product from the digestion (2878 base pairs) which contains the EF1a-puro-GFP cassette was saved for a later assembly reaction in the final cloning step (described below). The PCR amplified IDT oPool gRNAs with associated 10 N random barcodes were cloned into the linear backbone using NEBuilder HiFi DNA Assembly (NEB, Cat. No. E2621S) using 0.15 pmol of the insert (gRNA library) and 0.02 pmol of the linear backbone. Assembled product was transformed into electrocompetent cells (NEB, Cat. No. C3020K) and plasmid DNA was extracted with a midiprep kit (Zymo Research, Cat. No. D4200). The resulting vector was then digested with SapI (NEB, Cat. No. R0569S), for one hour at 37 degrees Celsius. Digested product was cleaned with 0.5X AMPure beads (Beckman Coulter, Cat. No. A63880) and cleaned digested linear backbone was used for a subsequent assembly reaction to add the EF1a-puro-GFP cassette back into the final piggyFlex vector between the gRNA sequence and the 10 N random barcode sequences. 0.014 pmol of the linear backbone was assembled with 0.056 pmol of the insert sequence and the assembly reaction was cleaned with a 0.5X AMPure step. The assembled product was transformed into electrocompetent cells and plasmid DNA was extracted with a midiprep kit. The final plasmid library was

subsequently PCR amplified and sequenced to ensure that all 493 gRNAs were successfully cloned into the piggyFlex vector. Note: The 10 N barcode is an additional gRNA identification strategy that can be used to assign gRNAs to cells, however, we used directly sequenced gRNAs (from the 10x Genomics capture sequence) to identify gRNAs in this work as this more accurately assigns gRNA transcripts to cells[38].

## Transfection of the gRNA library, selection, and cell culture

**K562 cells**. A total of 16 million K562 cells (8 million K562-VP64 cells and 8 million K562-VPR cells) were transfected with the gRNA library and the piggyBac transposase (System Biosciences, Cat. No. PB210PA-1) at a 20:1 molar ratio of library:transposase using a Lonza 4D nucleofector and the Lonza nucleofector protocol for K562 cells. The 16 million cells were split across 8100 uL nucleofection cartridges, with each individual nucleofection cartridge receiving two million cells and 2 ug of total DNA. After nucleofection, cells were transferred to pre-warmed RPMI media in a cell culture flask and incubated at 37 degrees Celsius. One day after transfection, cells were selected with 2 ug/mL puromycin (GIBCO/Thermo Fisher Scientific; Cat. No. A1113803). After 9 days, cells were harvested for single-cell transcriptome profiling.

## Induced pluripotent stem cells

Six million dCas9-VPH iPSCs (same cells as described above) were transfected with the gRNA library and the piggyBac transposase at a 5:1 molar ratio of library:transposase using the Lonza nucleofector and the Lonza nucleofector CB-150 program. The six million cells were split across 6100 uL nucleofection cartridges, with each individual nucleofection cartridge receiving one million cells and 17.5 ug of total DNA. After nucleofection, cells were transferred to pre-warmed mTeSr Plus basal medium with ROCK inhibitor in a cell culture flask and incubated at 37 degrees Celsius. One day after transfection, cells were selected with 20 ug/mL puromycin (note: the AAVS1-NGN2 construct has a puromycin resistance cassette on it, so a higher dose of puromycin was used to successfully select for cells that received a gRNA in the presence of an existing puromycin resistance cassette). Media changes were performed daily (mTeSr Plus basal medium with ROCK inhibitor and 10 ug/mL puromycin) for seven days prior to initiating neuron differentiation (described in "Human iPSC-derived neuronal cell culture, differentiation, and CRISPRa induction" methods section).

## 10x genomics sc-RNA-seq with associated gRNA transcript capture

**K562 screen**. Cells were harvested and prepared into single-cell suspensions following the 10x Genomics Single Cell Protocols Cell Preparation Guide (Manual part number CG00053, Rev C). Four lanes were used for the single-cell transcriptome profiling, with two lanes containing cells from the K562-VP64 cell line, and two lanes containing cells from the K562-VPR cell line. Roughly 10,000 cells were captured per lane of a 10× Chromium chip (Next GEM Chip G) using Chromium Next GEM Single Cell 3′ Reagent Kits v3.1 with Feature Barcoding technology for CRISPR Screening (10× Genomics, Inc, Document number CG000205, Rev D).

## iPSC-derived neuron screen

iPSC-derived neurons were harvested and prepared into single-cell suspensions following a published protocol[47]. Cells were split into two batches, with one batch going through a fluorescence-activated cell sorting (FACS) step to sort on the top 40% of green fluorescent protein (GFP) expression to enrich for neurons with greater numbers of gRNAs integrated, and the second batch going directly into the 10× Genomics single-cell library preparation protocol. Sorting on the top 40% of GFP expression resulted in a two-fold increase in the mean number of gRNAs integrated in those cells as compared to unsorted cells. Four lanes were used for the single-cell transcriptome profiling, with two lanes containing GFP-positive sorted cells, and two lanes containing

unsorted cells. Roughly 13,000 cells were captured per lane of a 10× Chromium high-throughput chip (Next GEM Chip M) using Chromium Next GEM Single Cell 3′ HT Reagent Kits v3.1 (Dual Index) with Feature Barcode technology for CRISPR Screening (10× Genomics, Inc, Document number CG000418, Rev C).

**Sequencing of scRNA-seq libraries**. Final libraries were sequenced on an Illumina NextSeq 2000 P3 100 cycle kit (R1:28 I1:10, I2:10, R2:90) for each screen (K562 and iPSC-derived neuron screens). Gene expression and gRNA transcript libraries were pooled at a 4:1 ratio for sequencing.

## Transcriptome data processing and quality control filtering for K562 and iPSC-derived neuron screens

CellRanger version 6.0.1 was used to perform bcl2fastq and count matrix generation. CellRanger mkfastq was run using default parameters, and CellRanger count was run using the GRCh38-3.0.0 reference transcriptome from 10× Genomics and default parameters. For the K562 screen, cells with greater than 10% mitochondrial reads and less than 4000 UMIs were filtered out. For the iPSC-derived neuron screen, cells with greater than 17% mitochondrial reads and less than 1500 unique molecular identifiers (UMIs) were filtered out. After quality control filtering, 33,944 cells were retained in the K562 screen, and 51,183 cells were retained in the iPSC-derived neuron screen. The resulting count matrix output after this filtering was used for all downstream analyses.

## Neuron differentiation transcriptome projection. Single-cell transcriptome data from a time course study of iPSC-derived neurons[28] was downloaded from https://www.ebi.ac.uk/biostudies/arrayexpress/studies/E-MTAB-10632 (Accession No. E-MTAB-10632, matrices_timecourse.tar.gz), and integrated with the neuron CRISPRa screening dataset described here. Seurat v4 was used for all data analyses[48]. The CRISPRa dataset was randomly downsampled to 5000 cells for this analysis. Count matrices from both matrices were filtered to include only shared genes from the two datasets ($n = 14,777$ genes). SelectIntegrationFeatures() and FindIntegrationAnchors() were run using default parameters to identify anchors for integration. 20,606 anchors were found and 2953 anchors were retained for data integration. IntegrateData() was run using the retained 2953 anchors to integrate the two datasets. After integration, standard Seurat single-cell analysis was performed to scale the data, and run the PCA and UMAP algorithms.

## gRNA assignment and differential gene expression testing

Genomic coordinates (hg38) for final gRNA spacers were isolated using a loop built around the matchPattern() function from the BSgenome package[49]. A 2 Mb window (1 Mb upstream and downstream) around each gRNA was then calculated and all genes within the 2 Mb window were isolated using a loop built around ENSEMBL biomaRt getBM() function[50,51]. These 1 Mb neighboring gene sets were then filtered to unique entries (unique HGNC symbols) for compatibility with the Seurat FindMarkers() function used in DE testing.

A global UMI filter of 5 gRNA UMIs/cell was used to assign gRNAs to single cell transcriptomes for both K562 and iPSC-derived neuron datasets (note this heuristic threshold was chosen based on manual inspection of the UMI count distributions for each gRNA and prior work)[2]. gRNA UMI counts for each cell were derived from the count matrix of passing cells output by CellRanger (which applies an automatic total UMI threshold to cells) and which also passed QC.

In our original approach, expression of a given gene within 1 Mb of the gRNA of interest was compared between all cells with a given gRNA and all other cells as control. log2() fold changes in expression for a given gene were calculated using the Seurat FindMarkers() function with the following arguments: ident.1 = gRNA_Cells, ident. 2 = Control_Cells, min.pct = 0, min.cells.feature = 0, min.cells.group = 0,

features = target_gene, logfc.threshold = 0. A Wilcoxon rank-sum test was used to generate raw differential expression *P*-values. This process was then iterated for all genes within 1 Mb of all gRNAs. NTCs were tested against all genes within 1 Mb of any targeting gRNA. Only tests involving genes detected in >0.2% of test gRNA and control cells were carried forward.

These raw differential expression *P*-values were then used to calculate empirical *P*-values to call EFDR < 0.1 sets[2]. Specifically, an empirical *P*-value was calculated for each gRNA-gene test as:

$$[(the\ number\ of\ NTCs\ with\ a\ P-value\ lower\ than\ that\ test's\ raw\ P-value)+1]/$$
$$[the\ total\ number\ of\ NTC\ tests+1]$$

Empirical *P*-values were then Benjamini-Hochberg corrected, and those <0.1 were kept for 10% EFDR sets.

With SCEPTRE, count matrices and associated metadata were used to construct a single cell covariate matrix with gRNA and gene expression library size (total UMIs), unique genes (non-zero expression in gRNA and gene expression libraries), cell line (in the case of K562 cells with differing CRISPRa effectors), whether cells were GFP sorted prior to profiling, 10X lane, and percent mitochondrial reads input as covariates during model fitting. A global UMI filter of 5 gRNA UMIs per cell was used to assign gRNAs to single cell transcriptomes. The run_sceptre_highmoi_experimental() function was used to run calibration (NTC) and discovery (tests) with default parameters (e.g. two-tailed, >7 targeting and control cells with non-zero expression). Each individual gRNA was tested against all neighboring genes within 1 Mb. NTCs were tested against all genes within 1 Mb of a targeting gRNA. NTCs were randomly downsampled to match the number of targeting cis tests for visualization on QQ plots. Resulting SCEPTRE *P*-values were then Benjamini-Hochberg corrected and those <0.1 were kept for two-sided discovery sets.

Log2 fold changes between gRNA and control cells were visualized using the gviz package[52] along with tracks for RefSeq transcripts (ENSEMBL biomaRt), H3K27ac, and ATAC seq peaks. The K562 ATAC and H3K27ac data were downloaded from ENCODE[53]. ATAC-seq and H3K27ac CUT&RUN data from 7–8 week old NGN2-iPSC inducible excitatory neurons was obtained from Song et al. [54]. As previously described, ATAC-seq and CUT&RUN reads were trimmed to 50 bp using TrimGalore with the command –hardtrim5 50 before alignment (https://github.com/FelixKrueger/TrimGalore). ATAC-seq reads were realigned to hg38 using the standard Encode Consortium ATAC-seq and ChIP-seq pipelines respectively with default settings and pseudo replicate generation turned off. Trimmed, sorted, duplicate and chrM removed ATAC-seq bam files from multiple biological replicates were combined into a single bam file using samtools merge v1.10[55]. Trimmed CUT&RUN reads were realigned to hg38 using Bowtie2 v2.3.5.1 with the following settings –local –very-sensitive-local –no-mixed –no-discordant -I 10 -X 700 and output sam files were convert to bam format using samtools view[55,56]. Duplicated reads were removed from the CUT&RUN bam file using Picard MarkDuplicates v2.26.0 with the –REMOVE_DUPLICATES =true and –ASSUME_SORTED=true options (http://broadinstitute.github.io/picard/). Finally, bam files were converted using the bedtools genomecov followed by the UCSC bed-GraphToBigWig utility.

For correlations of epigenetic features with CRISPRa hits, epigenetic feature datasets were downloaded from ENCODE or corresponding primary publications (datasets listed in Supplementary Data 8). Whether a gRNA was a SCEPTRE hit (Benjamini-Hochberg–adjusted SCEPTRE *P*-value < 0.1) was treated as a categorical input variable for Spearman correlations with epigenetic features quantified in 100 bp windows at the gRNA target sites. Similarly, various gRNA metrics (e.g. GC content) or gene level metrics (e.g. baseline expression) were previously quantified, either as part of the gRNA design process or gRNA assignment and differential expression testing

frameworks (described above) and tested for association with hit gRNAs.

## Statistics and reproducibility

No statistical method was used to predetermine sample size. The experiments were not randomized though by design allocation of gRNAs to individual cells is not programmed. The Investigators were not blinded to allocation during experiments and outcome assessment. Detailed statistical tests and quantitative treatment of data are otherwise described in the relevant Results or Methods sections above. No data were excluded from the analyses.

## Singleton replication and validations with bulk RNA-sequencing

To replicate and validate a gRNA's ability to upregulate its target gene outside of the pooled screening format, we generated individual, polyclonal cell lines that each expressed a single gRNA from a chosen representative set of gRNAs (Supplementary Data 15). This representative gRNA set was chosen to validate gRNAs that upregulate their targets in a cell-type specific manner (TBR1 is neuron-specific; ASIC1 and ANXA1 are enhancer targeting gRNAs that are K562-specific) and to validate promoter targeting gRNAs that are able to upregulate their targets in both cell types (e.g. BCL11A, TCF4, and FOXP1). gRNAs were ordered in two Oligo Pools (Integrated DNA Technologies), with each gRNA bearing a unique handle sequence that was used to uniquely PCR amplify out individual gRNA oligos (Supplementary Data 15). gRNAs were individually cloned into the piggyFlex vector exactly as described in the "gRNA library cloning into piggyFlex vector" methods section above, with the only difference being that the 10 N barcode was not included in the oligo design for these validation oligos. After cloning, the gRNA sequences in the piggyFlex vectors were sequence verified to ensure correct insertion of the gRNA sequences.

PiggyFlex constructs containing gRNA sequences were transfected into K562 cells and iPSCs with the Lonza 4D nucleofector (using the K562 protocol for K562 cells, and the CB-150 program for the iPSCs). 525 ng of total DNA (using a 5:1 molar ratio of piggyFlex vector and transposase vector) was transfected into three replicates of 30,000 cells each. After transfection, cells were seeded into 48-well plates for culture. One day later, 2 ug/mL (for K562 cells) and 20 ug/mL (for iPSCs) puromycin (GIBCO/Thermo Fisher Scientific; Cat. No. A1113803) was added to the cultures to select for successfully transfected cells. Daily media changes were performed (with puromycin) to continually select cells for seven days. After seven days of selection, K562 cells were lysed with TRIzol (Invitrogen, Cat. No. 15596026). Samples were frozen in TRIzol reagent at −20C, and RNA was extracted using the Direct-zol RNA miniprep kit (Zymo Research, Cat. No. R2050). After puromycin selection, iPSCs were differentiated into neurons exactly as described in the "Human iPSC-derived neuronal cell culture, differentiation, and CRISPRa induction" section. iPSC-derived neurons were differentiated for eight days prior to TRIzol lysis and RNA extraction as described above for K562 cells.

RNA-sequencing libraries were prepared using an in-house library preparation method. 30–50 ng of RNA was used as input for each library. RNA was reverse transcribed using indexed oligo dT primers (ACGACGCTCTTCCGATCTNNNNNNNNNAGAGAACTTGTTTTTTTTTTTTTTTTTTTTTTTTTTTTTTTTTTVN), template switching oligo (AAGCAGTGGTATCAACGCAGAGTGAATGGG), and Template Switching RT Enzyme Mix protocol from NEB (New Engand Biolabs, Cat. No. #M0466L). cDNA was PCR amplified using the NEBNext High-Fidelity 2X PCR Master Mix (New England Biolabs, Cat. No. M0541) with the following forward and reverse primers: ACGACGCTCTTCCGATC (forward), AAGCAGTGGTATCAACGCA (reverse). SYBR Green (Thermo Fisher Scientific, Cat. No. S7567) was added to track the amplification curve and proper PCR cycle number. PCR amplified cDNA was purified with 1X AMPure beads (Beckman Coulter, Cat. No. A63882), washed twice with 70% ethanol, eluted in 20 uL of nuclease-free water and loaded on

an Agilent Tapestation using D5000 reagents (Agilent Technologies, Cat. No. 5067-5589) and the corresponding screentape (Agilent Technologies, Cat. No. 5067-5588). 30–50 ng of purified PCR amplified cDNA was fragmented using an i7 loaded Tn5 transposase similar to the process described by Cao et al.[41]. To load the transposase, we followed the manufacturer's protocol to load unloaded transposase (Diagenode, Cat. No. C01070010-20). Tagmented DNA was PCR amplified using NEBNext High-Fidelity 2X PCR Master Mix (New England Biolabs, Cat. No. M0541) with indexed primers (CAAGCAGAA-GACGGCATACGAGATNNNNNNNNNNGTCTCGTGGGCTCGG and AATGATACGGCGACCACCGAGATCTACACNNNNNNNNNNA-CACTCTTTCCCTACACGACGCTCTTCCGATCT) following previously described PCR conditions by Cao et al.[41]. This PCR adds sample-specific indices and P5 and P7 Illumina sequencing adapters. Libraries were cleaned using a 1X AMPure bead cleanup (Beckman Coulter, Cat. No. A63882), and quantified on an Agilent TapeStation using D1000 reagents (Agilent, Cat. No. 5067-5583) and the corresponding screen-tape (Agilent, Cat. No. 5067-5582). For sequencing, libraries were diluted to 2 nM and pooled equimolarly, and then diluted to a loading concentration of 650 pM. Libraries were sequenced on an Illumina NextSeq 2000 using P1 100 cycle kits (R1: 18, I1:10, I2: 10 (optional), and R2:110). Transcript and gene-level quantifications were performed using kallisto[57]. Prior to transcript quantification, reads were down-sampled to the minimum number of reads observed for a sample to control for differences in sequencing depth. DESeq2[58] was used to quantify expression changes for all genes within 1 Mb of the gRNA target site and genome wide. Genome-wide differential analyses were conducted using default parameters. Target gene expression was compared between gRNA cells and control cells using a two-tailed Wilcoxon rank-sum test with a significance threshold of 0.1 (mirroring single-cell analysis framework described above). Gviz was used to visualize differential expression testing results alongside tracks for neighboring genes (ENSEMBL biomaRt) as described above.

### Reporting summary

Further information on research design is available in the Nature Portfolio Reporting Summary linked to this article.

## Data availability

Raw sequencing data have been uploaded to the Sequence Read Archive with associated BioProject ID PRJNA1157910. Additionally, the raw data, processed data, and corresponding metadata are have been deposited to the IGVF database under accession codes IGVFDS9078ZWQH and IGVFDS4021XJLW. These are also available at[https://krishna.gs.washington.edu/content/members/CRISPRa_QTL_website/public/]. Published data used: GSE170378, GSM733656, GSE113483. Source data are provided with this paper.

## Code availability

All code and scripts used for analyses are all publicly available and are accessible on Github via the following link [https://github.com/shendurelab/multiplex_scCRISPRa_screening] or at [https://krishna.gs.washington.edu/content/members/CRISPRa_QTL_website/public/].

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

## Acknowledgements

We are grateful to members of the Shendure and Ahituv labs, as well as members of the Sanders, Bender, Bateup, and Feldman labs for comments, suggestions, and discussions on this work. We are particularly grateful to the Shendure lab gene regulation subgroup for technical advice and deep discussions regarding the development of the CRISPRa screening method. We would also like to thank Haedong Kim who piloted the in-house bulk RNA-seq protocol and provided reagents for the bulk RNA-seq experiments. The Human WTC11 NGN2 ecDHFR-dCas9-VPH line was a kind gift from the M. Kampmann lab at UCSF. Lenti dCAS9-VP64_GFP (Addgene plasmid # 61422) was a kind gift from the F. Zhang lab at Broad/MIT. This work was supported by the Weill Neurohub (to S.J.S., N.A., and J.S.), the National Human Genome Research Institute (UM1HG011966 to N.A. and J.S.) and the National Institute of Mental Health (U01MH122681 and R01MH116999 to S.J.S.). T.A.M. was supported by a Banting Postdoctoral Fellowship from the Natural Sciences and Engineering Research Council of Canada (NSERC). N.F.P. was supported by a National Science Foundation (NSF) graduate research fellowship. J.B.L. is a Fellow of the Damon Runyon Cancer Research Foundation (DRG-2435-21). D.C. was supported by award no. F32HG011817 from the National Human Genome Research Institute. J.S. is an Investigator of the Howard Hughes Medical Institute.

## Author contributions

Conceptualization, J.S. and N.A.; investigation, F.M.C., T.A.M., N.F.P., and R.M.D.; data curation, F.M.C., T.A.M., N.F.P, and R.M.D.; formal analysis, F.M.C., T.A.M., and N.F.P.; visualization, F.M.C. and T.A.M.; resources, N.A. and J.S.; supervision, L.S., S.J.S., N.A. and J.S.; writing—original draft, F.M.C., T.A.M., J.S.; writing—review & editing, F.M.C., T.A.M., N.F.P., B.M., S.D., S.R., J.B.L., D.C., R.M.D., X.L., L.S., S.J.S., N.A and J.S.; funding acquisition, S.J.S., N.A., and J.S.

## Competing interests

S.J.S. receives research funding from BioMarin Pharmaceutical Incorporated. N.A. is the cofounder and on the scientific advisory board of Regel Therapeutics and receives funding from BioMarin Pharmaceutical Incorporated. J.S. is a scientific advisory board member, consultant and/or co-founder of Cajal Neuroscience, Guardant Health, Maze Therapeutics, Camp4 Therapeutics, Phase Genomics, Adaptive

Biotechnologies, Scale Biosciences, Sixth Street Capital, Prime Medicine, Somite Therapeutics and Pacific Biosciences. All other authors declare no competing interests.
