## [Peer Review File · Nature Communications]

Multiplex, single-cell CRISPRa screening for cell type specific regulatory elementsEditorial Note: This manuscript has been previously reviewed at another journal that is not operating a transparent peer review scheme. This document only contains reviewer comments and rebuttal letters for versions considered at *Nature Communications*.

Reviewers' comments:

Reviewer #1 (Remarks to the Author):

I thank the authors for adding the genome-wide analysis of the validation RNA-seq data. I agree this further validates some of their hits.

I note there seems to me to be a typo (in the rebuttal document only), wherein BCL11A is mentioned as both an example of a genome-wide significantly upregulated target (correctly) and as a not-significantly upregulated target (incorrectly, I believe).

I recommend the paper be accepted and commend the authors on their work. In particular, I expect the piggyBac and iPSC-neuron CRISPRa methodologies and the functionally validated enhancer-gene links will be directly useful to other researchers. In addition, the authors' scRNA-seq approach is well-suited to quantify CRISPRa robustness across many targets, and I think the results should speak for themselves (without being hyped or being deemed not good enough for publication). In my view, this paper is a necessary contribution to the literature, wherein CRISPRa's success rates are not very well documented.

Reviewer #3 (Remarks to the Author):

While I appreciate the author's detailed response, I remain unconvinced that this manuscript provides a major conceptual or technical advance. In their response to my concerns and novelty concerns of Reviewer #1, the authors highlight their novel "framework" for sc-CRISPRa screens in post-mitotic human cell types and its utility to investigate human enhancers. However, as mentioned in my previous comments, the screen in iPSC-derived neurons appears to be of very limited quality: Out of a complex library, the authors identify only 17 scoring sgRNAs (35 according to SPECTRE analysis), among which 6 were positive controls and not a single sgRNA targeting an enhancer (1 based on SPECTRE). Validation data are only provided for 8 sgRNAs targeting some of the positive controls and 4/9 disease genes. Overall, validated sgRNAs were only identified for 3 of the investigated disease genes (BCL11A, TCF4, TBR1). Effects sizes vary to a large degree and do not support the conclusion that 7/8 sgRNAs triggered a "consistent, clear upregulation" in iPSC derived neurons (besides FOXP1, ANXA1 and ASIC1 also look insignificant, with p-values of 0.2 and 0.8, respectively). Although I agree that 2-fold effects would be desired for haploinsufficient genes, there seems to be little control over effect sizes for individual sgRNAs. More generally, given that the screen yielded validated sgRNAs for only 3/9 candidate genes, I continue to find claims about therapeutic applications for "hundreds of disease genes" far-fetched. Overall, the screen in iPSC-derived neurons is very limited in scope, technical quality and success, and thus does not sufficiently support the major novelty claims of this study.

POINT-BY-POINT REBUTTAL

We thank the reviewers for their additional comments, and the editor for the opportunity to appeal the decision. Below we provide point-by-point responses to the latest comments of Reviewers #1 and #3. The reviewers' comments are in **blue** and our responses in **black**.

Per editorial guidance, we have not sought to address Reviewer #3's concerns about novelty, and we have also modified the manuscript to further tone down claims about therapeutic potential.

Reviewers' comments:

Reviewer #1 (Remarks to the Author):

I thank the authors for adding the genome-wide analysis of the validation RNA-seq data. I agree this further validates some of their hits.

We thank the reviewer for this positive comment, and for agreeing that the new RNA-seq data further validates some of the hits.

I note there seems to me to be a typo (in the rebuttal document only), wherein BCL11A is mentioned as both an example of a genome-wide significantly upregulated target (correctly) and as a not-significantly upregulated target (incorrectly, I believe).

Thank you for catching this. The corrected text now reads as follows:

“In all other cases, the primary target was not genome-wide significant. In some of these cases, there were also no DEGs genome-wide (e.g. HMGA1 in either cell line), while in yet other cases (e.g. TCF4 in K562 cells, and TBR1 in iPSC-derived neurons), secondary DEGs were detected despite the primary target not achieving genome-wide significance.”

I recommend the paper be accepted and commend the authors on their work. In particular, I expect the piggyBac and iPSC-neuron CRISPRa methodologies and the functionally validated enhancer-gene links will be directly useful to other researchers. In addition, the authors' scRNA-seq approach is well-suited to quantify CRISPRa robustness across many targets, and I think the results should speak for themselves (without being hyped or being deemed not good enough for publication). In my view, this paper is a necessary contribution to the literature, wherein CRISPRa's success rates are not very well documented.

We thank the reviewer for this recommendation and commendation. We agree that the piggyBAC approach will be directly useful as a powerful alternative to lentiviral strategies that do not work well in human iPSCs, which to date has sharply constrained the range of *in vitro* human models in which single-cell CRISPR screens can be performed (largely restricting them to workhorse cancer cell lines). We also agree that the scRNA-seq approach is well-suited to advance the utility and robustness of CRISPRa for discovering enhancer-gene links and other purposes. We appreciate the characterization of the paper as a “necessary contribution” to the literature, where indeed there is very little information about CRISPRa at scale.

Reviewer #3 (Remarks to the Author):

While I appreciate the author's detailed response, I remain unconvinced that this manuscript provides a major conceptual or technical advance.

Per editorial guidance surrounding the submission of this appeal, we have been instructed that the concern over novelty does not need to be further addressed.

In their response to my concerns and novelty concerns of Reviewer #1, the authors highlight their novel "framework" for sc-CRISPRa screens in post-mitotic human cell types and its utility to investigate human enhancers. However, as mentioned in my previous comments, the screen in iPSC-derived neurons appears to be of very limited quality: Out of a complex library, the authors identify only 17 scoring sgRNAs (35 according to SPECTRE analysis), among which 6 were positive controls and not a single sgRNA targeting an enhancer (1 based on SPECTRE).

The reviewer seems to have missed a key point, which is that **the candidate enhancers that were tested in both K562 cells and iPSC-derived neurons were K562 candidate enhancers**. Therefore we do not necessarily expect them to drive upregulation when targeted by CRISPRa in iPSC-derived neurons. This is central to our main findings, i.e. that our framework is able to identify distal elements that act in a cell type-specific manner when targeted by CRISPRa. To reiterate, the fact that not a single sgRNA targeting a **K562** candidate enhancer was a hit in iPSC-derived neurons does not undermine the quality of the findings. Whether enhancers for a specific cell type (here, K562 cells) can be activated by CRISPRa to drive expression in some other cell type (here, iPSC-derived neurons) was an open question at the outset of the study, and our data suggest that no, they cannot be. The result is negative, but answers the question that the experiment set out to ask.

The reviewer also appears focused on the contrast between the complexity of the screen (493 targets) and the number of hits (83 in the K562 screen and 35 in the iPSC-derived neuron screen). It is not clear to us what hit rate the reviewer would have expected. These "hit rates" (17%, 7%) are on par if not higher than other studies that leverage CRISPR to query non-coding regions. For example, a recent meta-analysis of 108 noncoding CRISPR screens in human cell lines¹, comprising >540,000 CRISPR perturbations across nearly 25 megabases of the genome (overwhelmingly CRISPRi) confirmed only 332 CRE-gene links in K562 cells¹ ($332/540,000 = 0.06\%$). With the caveat that many of the contributing studies were tiling screens, our hit rate is on the order of two orders of magnitude higher than observed in this meta-analysis summarizing the current state of the field.

Looking at this same recent meta-analysis¹, **none of the >100 published screens contributing data are in iPSC-derived human cell types** (a few accessions are from undifferentiated WTC11 iPSCs, which would not have manifested the lentivirus-associated silencing that occurs during differentiation and led us to develop the piggyBAC alternative reported in this paper). Also, note that **only one of 123 contributing accessions** is a CRISPRa screen (as opposed to CRISPRi, CRISPR-cut, or dCas9), and as the authors say, CRISPRa "has not yet been as widely adopted for noncoding screens, and more data are needed to inform guidelines for its use."

Validation data are only provided for 8 sgRNAs targeting some of the positive controls and 4/9 disease genes. Overall, validated sgRNAs were only identified for 3 of the investigated disease genes (BCL11A,

TCF4, TBR1).

It's unclear to us what the reviewer means when they say that we are targeting positive controls.

To the broader point, the singleton validations are experimentally laborious, and we felt that 8 was an entirely reasonable number to validate, especially given how we selected these 8 sgRNAs. To be clear, the purpose of the validations were not to validate specific gRNAs for therapeutic purposes, but rather to validate the results of the screen itself, so we were not focused on getting one validated gRNA per target gene. We certainly could have done so, but we would argue that would have been misleading. There is a tendency in the genomics field (including this subfield) to perform validations on the strongest hits coming out of screens, but this risks artifactually inflating confidence in the screen as a whole. We made a conscious choice not to do this here, and we still think that it was the right choice. As clearly stated in the revision, the 8 hit gRNAs were selected "to represent a range of significance levels, as well as both shared and cell-type specific promoters and enhancers". **Figs. 4a** and **4c** were specifically designed to highlight this representativeness. The validation of these 8 gRNAs required cloning them as singleton constructs, introducing them at high MOI to create 3 independent polyclonal cell lines in each cell context, in the case of neurons differentiating them, and then finally creating bulk RNA-seq libraries for each line. Given 8 gRNAs x 2 cell contexts x 3 replicate cell lines, we essentially constructed 48 independent polyclonal lines (24 of which were differentiated to neurons) and performed bulk RNA-seq samples from these. This was a lot of work, and given that it's a representative set, it's not clear what value we would have gained from doubling or tripling its size.

We also point out that the validation rate is not referencing whether an sgRNA is a hit or not, but rather whether it is behaving as predicted from the screen. Of the 8 sgRNAs selected for the validations, 7 were expected to drive upregulation in K562 cells based on the initial screen, and 6 did (all but the sgRNA targeting *HMGA1*). Of the 8 sgRNAs selected for the validations, 6 were expected to drive upregulation in iPSC-derived neurons based on the initial screen, and 5 did (all but the sgRNA targeting *FOXP1*). These "hit" validation rates (6/7; 5/6) are entirely consistent with the 0.1 FDR. Overall, we were pleased by the high validation rate from this **representative set** of gRNAs, and stand by our choices.

Effects sizes vary to a large degree and do not support the conclusion that 7/8 sgRNAs triggered a "consistent, clear upregulation" in iPSC derived neurons (besides *FOXP1*, *ANXA1* and *ASIC1* also look insignificant, with p-values of 0.2 and 0.8, respectively).

The context of this comment leads us to believe that the reviewer is referencing differences in effect sizes for the validation set of 8 sgRNAs in terms of how they performed in K562 cells (**Fig. 4b**) vs. iPSC-derived neurons (**Fig. 4d**). We believe that the reviewer is missing a key point, which is that **not every gRNA validated was a hit in the initial screen in both cell types**. In particular, the *ANXA1* and *ASIC1* sgRNAs were only a hit in the K562 screen. The varying effect sizes in K562 cells vs. iPSC-derived neurons is predicted and expected. As we state in the text:

"Hit gRNAs targeting e-*ANXA1* and e-*ASIC1*, two enhancers identified as K562-specific in our single-cell analysis, drove upregulation of *ANXA1* and *ASIC1* exclusively in K562 cells in singleton validations."

Thus, the cited p-values of 0.2 and 0.8 correspond to the expected **lack** of activity for these gRNAs in iPSC-derived neurons. Although this comment leaves us worried that the reviewer is not reading the

revision carefully, how we introduced these points could have been clearer. We therefore revised the opening of the paragraph summarizing validation results to read:

“Our validation set included sgRNA hits that are expected to drive upregulation in both cell types (*BCL11A*, *DNMT3B*, *TCF4*, *FOXP1*, *HMGA1*), in K562 cells only (*ANXA1*, *ASIC1*), or in iPSC-derived neurons only (*TBR1*). In 6 of 7 (K562) or 5 of 6 cases (iPSC-derived neurons), the expected upregulation was observed in the expected cell type, validation rates consistent with the 0.1 FDR of the initial screens (Fig. 4; Fig. S20).”

Although I agree that 2-fold effects would be desired for haploinsufficient genes, there seems to be little control over effect sizes for individual sgRNAs.

We are glad the reviewer understands the case for why 2-fold effects are appropriate for CRISPRa of haploinsufficient genes. Our current inability to predict exactly which sgRNAs will be effective, nor the precise magnitude of their effect sizes, are precisely why we need a method that facilitates scalable empirical screens.

More generally, given that the screen yielded validated sgRNAs for only 3/9 candidate genes, I continue to find claims about therapeutic applications for “hundreds of disease genes” far-fetched.

Per this comment and editorial guidance, we have gone through the manuscript and almost entirely removed claims related to therapeutic potential. We still refer to CRT in the introduction (to contextualize the importance of CRISPRa as a modality) and bring up therapeutic possibilities once in the discussion (to contextualize the ideal effect size for treating haploinsufficient disorders), but other claims have been deleted, and the final paragraph that includes the quoted “hundreds of disease genes” has been deleted in its entirety.

Overall, the screen in iPSC-derived neurons is very limited in scope, technical quality and success, and thus does not sufficiently support the major novelty claims of this study.

Per editorial guidance surrounding the submission of this appeal, we have been instructed that the concern over novelty does not need to be further addressed.

References

1. Yao, D. *et al.* Multicenter integrated analysis of noncoding CRISPRi screens. *Nat. Methods* **21**, 723–734 (2024).

REVIEWERS' COMMENTS

Reviewer #4 (Remarks to the Author):

The manuscript by Chardon and colleagues provides insights that will be helpful to those attempting to use CRISPRa to target genes of interest and to understand the regulatory logic of CREs. While I appreciate the suggestions of Reviewer 3 to reign in claims of therapeutic relevance, which is a very very long road, I see clear value in the manuscript as-is, and recommend publication.

My only suggestion is to include a citation to PMID: 36917981, which provides additional context for the challenges of engineering cells with CRISPRa machinery.

Reviewer #4 (Remarks to the Author):

The manuscript by Chardon and colleagues provides insights that will be helpful to those attempting to use CRISPRa to target genes of interest and to understand the regulatory logic of CREs. While I appreciate the suggestions of Reviewer 3 to reign in claims of therapeutic relevance, which is a very very long road, I see clear value in the manuscript as-is, and recommend publication.

My only suggestion is to include a citation to PMID: 36917981, which provides additional context for the challenges of engineering cells with CRISPRa machinery.

We thank the reviewer for their positive assessment of our manuscript and for recommending publication. We have added the suggested reference (reproduced below).

“Indeed, delivery of CRISPRa machinery to post-mitotic neurons is considerably more challenging than workhorse cancer cell lines and requires more complex cell engineering and delivery approaches(Wu et al. 2023).”

References

Wu, Qianxin, Junjing Wu, Kaiser Karim, Xi Chen, Tengyao Wang, Sho Iwama, Stefania Carobbio, et al. 2023. “Massively Parallel Characterization of CRISPR Activator Efficacy in Human Induced Pluripotent Stem Cells and Neurons.” *Molecular Cell* 83 (7): 1125–39.e8.